# Intrinsic OXPHOS limitations underlie cellular bioenergetics in leukemia

Margaret AM Nelson[1,2†], Kelsey L McLaughlin[1,2†], James T Hagen[1,2], Hannah S Coalson[1,2], Cameron Schmidt[1,2], Miki Kassai[2,3], Kimberly A Kew[3], Joseph M McClung[1,2,4], P Darrell Neufer[2], Patricia Brophy[2], Nasreen A Vohra[5], Darla Liles[6], Myles C Cabot[2,3], Kelsey H Fisher-Wellman[1,2*]

[1]Department of Physiology, Brody School of Medicine, East Carolina University, Greenville, United States; [2]East Carolina Diabetes and Obesity Institute, East Carolina University, Greenville, United States; [3]Department of Biochemistry and Molecular Biology, Brody School of Medicine, East Carolina University, Greenville, United States; [4]Department of Cardiovascular Sciences, Brody School of Medicine, East Carolina University, Greenville, United States; [5]Department of Surgery, Brody School of Medicine, East Carolina University, Greenville, United States; [6]Department of Internal Medicine, Brody School of Medicine, East Carolina University, Greenville, United States

*For correspondence:
fisherwellmank17@ecu.edu

†These authors contributed equally to this work

Competing interests: The authors declare that no competing interests exist.

**Abstract** Currently there is great interest in targeting mitochondrial oxidative phosphorylation (OXPHOS) in cancer. However, notwithstanding the targeting of mutant dehydrogenases, nearly all hopeful 'mito-therapeutics' cannot discriminate cancerous from non-cancerous OXPHOS and thus suffer from a limited therapeutic index. Using acute myeloid leukemia (AML) as a model, herein, we leveraged an in-house diagnostic biochemical workflow to identify 'actionable' bioenergetic vulnerabilities intrinsic to cancerous mitochondria. Consistent with prior reports, AML growth and proliferation was associated with a hyper-metabolic phenotype which included increases in basal and maximal respiration. However, despite having nearly 2-fold more mitochondria per cell, clonally expanding hematopoietic stem cells, leukemic blasts, as well as chemoresistant AML were all consistently hallmarked by intrinsic OXPHOS limitations. Remarkably, by performing experiments across a physiological span of ATP free energy, we provide direct evidence that leukemic mitochondria are particularly poised to consume ATP. Relevant to AML biology, acute restoration of oxidative ATP synthesis proved highly cytotoxic to leukemic blasts, suggesting that active OXPHOS repression supports aggressive disease dissemination in AML. Together, these findings argue against ATP being the primary output of leukemic mitochondria and provide proof-of-principle that restoring, rather than disrupting, OXPHOS may represent an untapped therapeutic avenue for combatting hematological malignancy and chemoresistance.

## Introduction

In comparison to normal hematopoietic cells, various human leukemias present with increases in mitochondrial mass and higher basal respiration rates (*Farge et al., 2017*; *Goto et al., 2014b*; *Goto et al., 2014a*; *Sriskanthadevan et al., 2015*; *Suganuma et al., 2010*), the latter of which appears to sensitize them to global disruptions in mitochondrial flux (*Kuntz et al., 2017*; *Lagadinou et al., 2013*; *Lee et al., 2015*; *Mirali et al., 2020*; *Skrtić et al., 2011*; *Xiang et al., 2015*). Although these studies have ignited interest in mitochondrial-targeted chemotherapeutics (*Guièze et al., 2019*; *Panina et al., 2020*), experimental rationale for targeting OXPHOS in leukemia is largely based on the assumption that heightened respiration is representative of the cancerous

mitochondrial network's attempt to accommodate an increased ATP demand (i.e. increased 'OXPHOS reliance'). However, identical increases in mitochondrial respiration can derive from any number of physiological stimuli, ranging from increased demand for ATP resynthesis to decreased OXPHOS efficiency. Distinguishing between these potential outcomes is critical, as such insight likely demarcates targeted drug efficacy from undesirable systemic toxicity. For example, it is currently unclear how targeting increased 'OXPHOS reliance' in leukemia can specifically disrupt leukemic oxidative metabolism without impacting OXPHOS in other highly metabolic organs (e.g. brain, heart, muscle).

Given the ubiquitous necessity of OXPHOS for healthy cellular metabolism, a major barrier to mitochondrial-targeted drugs in leukemia relates to the need for cancer-cell selectivity (*Cohen, 2010*; *Dykens and Will, 2007*). The current project was based on the premise that establishing cause and effect between mitochondrial bioenergetics and cancer is one of the keys to developing targeted and more effective therapies. As a first step, this will undoubtedly require advanced technical approaches capable of quantifying the interplay among the major mitochondrial thermodynamic free energy driving forces to distinguish between changes in bioenergetic demand versus efficiency. To this end, our group recently developed a diagnostic biochemical workflow that quantifies the changes in free energy driving forces over the entire range of respiratory demand, thus providing a comprehensive profile of mitochondrial bioenergetic efficiency and capacity, relative to the underlying proteome (*Fisher-Wellman et al., 2019*; *Fisher-Wellman et al., 2018*; *McLaughlin et al., 2018*).

Herein, we leveraged our mitochondrial diagnostics workflow across several hematological cell types, including AML cell lines, primary human leukemias, and AML cells made refractory to the chemotherapeutic venetoclax. Clonal cell expansion in leukemia, including chemoresistant AML, was universally associated with two primary phenotypes; (1) higher basal respiration driven by increased cellular mitochondrial content, and (2) intrinsic OXPHOS repression. Parallel assessment of the underlying mitochondrial proteome linked this unique ability to bolster cellular mitochondrial content while simultaneously constraining oxidative ATP synthesis to shifts in adenine nucleotide translocase (ANT) isoform expression. Specifically, decreased ANT1 and increased ANT2/ANT3 appears to prime leukemic mitochondria for ATP uptake, rather than export, that, in turn directly inhibits the ability of mitochondrial OXPHOS to contribute to the cellular ATP-free energy ($\Delta G_{ATP}$). These findings are consistent with recent evidence demonstrating that mitochondrial OXPHOS is dispensable for tumor growth (*Martínez-Reyes et al., 2020*), and raise the intriguing possibility that the requirement for mitochondria in leukemia may have little to do with oxidative ATP production but instead reflect a requirement for unimpeded mitochondrial flux to support other aspects of anabolic growth [e.g., NAD$^+$ regeneration (*Luengo et al., 2020*), nucleotide synthesis (*Martínez-Reyes et al., 2020*)]. Critically however, accommodating such a flux demand, at least in leukemia, appears to require intrinsic mitochondrial remodeling that allows for forward electron transport/oxygen consumption to occur alongside ATP consumption. Given that acute restoration of OXPHOS kinetics in AML proved highly cytotoxic to leukemic blasts, the present findings provide proof-of-principle that interventions designed to restore, rather than disrupt, OXPHOS may impart therapeutic efficacy across various hematological malignancies.

## Results

### Mitochondrial bioenergetic profiling of acute leukemia reveals respiratory flux limitations

To begin to characterize the mitochondrial network in leukemia, we selected three commercially available acute leukemia cell lines – HL-60, KG-1, MV-4–11 – and comprehensively evaluated their bioenergetic profiles. These cells arise from unique precursors along the hematopoietic lineage, express a diverse array of cell surface markers, and have distinct underlying genetics (*Inoue et al., 2014*; *Mrózek et al., 2003*; *Rücker et al., 2006*). Results were compared to peripheral blood mononuclear cells (PBMC) isolated from healthy volunteers. The decision to use PBMC as a control was based on the assumption that comprehensive bioenergetic phenotyping of non-proliferative PBMC compared to various AML cell lines would provide sufficient experimental design contrast to reveal fundamental mitochondrial bioenergetic phenotypes potentially required for clonal cell expansion.

Using intact PBMC and leukemia cell lines, respiratory flux ($JO_2$) was assessed under basal conditions, as well as in response to ATP synthase inhibition (oligomycin), and FCCP titration (i.e. mitochondrial uncoupler). All experiments were performed in bicarbonate-free IMDM growth media, supplemented with 10% FBS. Following FCCP titration, respiration was inhibited with a combination of rotenone (inhibits complex I) and antimycin A (inhibits complex III). Consistent with prior work in human leukemia (*Jitschin et al., 2014*; *Sriskanthadevan et al., 2015*), basal respiration normalized to cell count was elevated above PBMC across all leukemia lines and maximal respiratory flux was higher in KG-1 and MV-4–11 (*Figure 1A*). When normalized to basal respiration, oligomycin similarly inhibited respiration across groups and the fold change induced by FCCP was consistently blunted in leukemia (*Figure 1B*).

Given the large differences in cell size between PBMC and leukemia (*Figure 1—figure supplement 1A*), we reasoned that normalization to total protein would likely provide the most accurate index of absolute respiratory kinetics across groups. Interestingly, upon normalization to total protein, although basal respiration remained higher in KG-1 and MV-4–11, differences in maximal respiratory flux were eliminated, particularly at higher FCCP concentrations (*Figure 1C*). In fact, maximal FCCP-supported $JO_2$ was nearly two-fold lower in HL-60 compared to PBMC when normalized to total protein (*Figure 1C*). Relative to PBMC, maximal respiration induced by FCCP occurred at much lower concentrations in leukemia (i.e. lower Km; *Figure 1D*), with increasing FCCP concentrations leading to an overt bioenergetic collapse (i.e. diminishing respiration rates; *Figure 1C*; compare $JO_2$ at FC [2.0 μM] vs FC [5.0 μM]). Similar findings were observed using the mitochondrial uncoupler BAM15 (*Figure 1—figure supplement 1B–C*). Measurements of extracellular acidification (ECAR), an indirect readout of glycolytic flux, revealed higher extracellular acidification and a rightward shift in ECAR relative to oxygen consumption rate (OCR) in AML cell lines, consistent with prior reports detailing a hyper-metabolic phenotype in leukemia (*Figure 1—figure supplement 1E–F*; *Suganuma et al., 2010*).

To determine if flux differences in leukemia could be explained by differences in mitochondrial content, nuclear and mitochondrial volumes were assessed independently by tetramethylrhodamine methyl ester (TMRM) or MitoTracker fluorescence and confocal microscopy (*Figure 1E–I*, *Figure 1—figure supplement 1D*). Absolute nuclear and mitochondrial volumes were higher in all leukemia lines (*Figure 1E–H*), consistent with leukemia's larger cell size (*Figure 1—figure supplement 1A*). However, when normalized to nuclear volume, mitochondrial content was elevated above PBMC only in HL-60 and MV-4–11 (*Figure 1I*). Interestingly, across all cell types, considerable discrepancies were apparent when protein-normalized maximal respiratory flux (*Figure 1C*) was compared to mitochondrial content (*Figure 1I*). This was particularly evident in HL-60 cells where mean maximal respiration, relative to the size of the underlying mitochondrial network, was ~5-fold lower compared to PBMC (*Figure 1J*). Together, these data suggested that at least a portion of the expansive mitochondrial network in AML may be biochemically constrained and thus unable to contribute to oxidative metabolism under basal conditions.

## Intrinsic limitations to OXPHOS characterize the mitochondrial network in leukemia cell lines

To directly test OXPHOS kinetics in leukemia, two complementary assays were designed. Both assays used digitonin-permeabilized cells energized with identical carbon substrates, and respiratory flux was stimulated with either FCCP or ATP free energy. In the first assay, the maximal capacity of the electron transport system (ETS) was assessed by energizing permeabilized cells with saturating carbon substrates (P/M/G/S/O) and titrating in FCCP (*Figure 2A–B*). The use of multiple substrates was intended to saturate carbon substrate availability such that maximal ETS flux could be quantified. Using this approach, absolute respiration in substrate-replete permeabilized cells was comparable to that observed using intact cells treated with FCCP (*Figure 2C*), confirming maximal ETS flux in the permeabilized system. Note, maximum FCCP-supported flux under these conditions is indicated throughout as $JH^+_{Total}$ (*Figure 2A*). Relative to PBMC, $JH^+_{Total}$ was lower in HL-60, unchanged in KG-1, and higher in MV-4–11 (*Figure 2B*).

In mammalian cells, the vast majority of the adenylate pool is represented by ATP (i.e., $\Delta G_{ATP}$), with typical values for ATP-free energy ranging from −56 to −64 kJ/mol (*Luptak et al., 2018*; *Roth and Weiner, 1991*; *Veech et al., 2002*). Thus, to evaluate OXPHOS kinetic efficiency in leukemia across a physiological range of ATP resynthesis demands, we utilized the creatine kinase (CK)

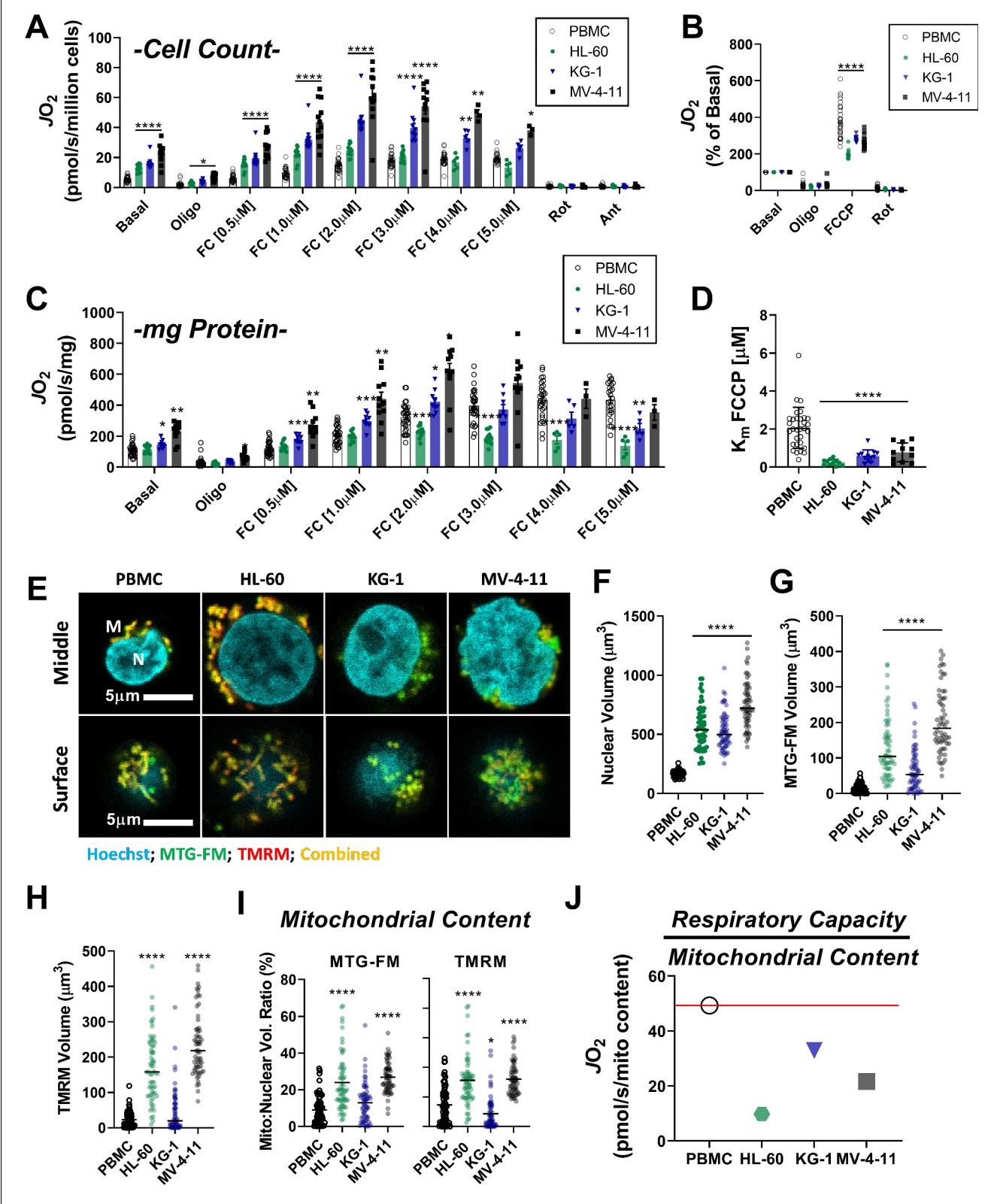

**Figure 1.** Leukemia exhibits impaired cellular respiratory capacity amid an increased mitochondrial network. All experiments were performed in intact cells. FCCP-stimulated flux normalized to cell count (A) and protein concentration (C) and represented as percentage of basal respiration (B). (D) $K_m$ of FCCP calculated from FCCP titration (cell lines n = 12, PBMC n = 31). (E-I) Confocal microscopy was performed using two mitochondrial targeted cationic fluorescent dyes, MitoTracker Green FM (MTG-FM) and TMRM. (E) Representative fluorescent images of nuclear volume (F) and mitochondrial

*Figure 1 continued on next page*

*Figure 1 continued*

volume as measured by MTG-FM (**G**) and TMRM labeling (**H**) (n = 65 cells/cell type). (**I**) Ratios of mitochondrial to nuclear volumes assessed by MTG-FM and TMRM labeling. (**J**) Respiratory deficiency of cell type calculated by comparing respiratory capacity (the protein-normalized maximal respiration rate) to mitochondrial content (mitochondrial TMRM volume). Data are presented as mean ± SEM and analyzed by two-way ANOVA (**A-D**) and one-way ANOVA (**F-I**). *p<0.05, **p<0.01, ***p<0.001, ****p<0.0001.

The online version of this article includes the following source data and figure supplement(s) for figure 1:

**Source data 1.** Raw values for '*Figure 1*' and '*Figure 1—figure supplement 1*'.

**Figure supplement 1.** Morphology of leukemia and respiratory flux stimulated by BAM15, a mitochondrial uncoupler.

energetic clamp (*Fisher-Wellman et al., 2018*; *Glancy et al., 2013*; *Messer et al., 2004*). This technique leverages the enzymatic activity of CK, which couples the interconversion of ATP and ADP to that of phosphocreatine (PCR) and free creatine (CR) such that extramitochondrial ATP-free energy (i.e. $\Delta G_{ATP}$) can be empirically titrated using PCR. Using this approach, permeabilized cells were energized with the same carbon substrate mix used for the ETS capacity assay and respiration was stimulated at minimal ATP-free energy. Note, $\Delta G_{ATP}$ equal to $-54.16$ kJ/mol reflects an ATP/ADP ratio in vivo that would be expected to induce 'maximal' OXPHOS flux and is thus referred to throughout as '$JH^+_{OXPHOS}$' (*Figure 2D*). Cytochrome C (Cyt C) was added to assess the integrity of the mitochondrial outer-membrane, and $\Delta G_{ATP}$ was then titrated via sequential additions of PCR. With respect to the ETS capacity assay, respiration stimulated by $\Delta G_{ATP}$ partially normalized $JO_2$ between MV-4–11 and PBMC and revealed decreased respiratory kinetics in both HL-60 and KG-1 (*Figure 2E*).

In both assays, the utilization of identical substrates ('P/M/G/S/O') allowed us to directly quantitate absolute OXPHOS kinetics ('$JH^+_{OXPHOS}$'), relative to the maximal capacity of the electron transport system ('$JH^+_{Total}$'). Together, $JH^+_{OXPHOS}$ and $JH^+_{Total}$ provide a quantitative index of fractional OXPHOS capacity as the ratio of the two reflects the proportion of the entire respiratory system that can be used for OXPHOS (*Figure 2F*). A ratio of '1' reflects maximal OXPHOS reliance, whereas a ratio of '0' indicates that the mitochondrial proton current cannot be utilized for ATP synthesis. Strikingly, calculated fractional OXPHOS in leukemic mitochondria was consistently decreased compared to PBMC, corresponding to a factor of ~0.5 (*Figure 2G*), indicating that only half of the available ETS capacity in leukemia can be dedicated to OXPHOS under physiological ATP free energy constraints. Given that the OXPHOS network is responsible for driving ATP/ADP disequilibrium to establish cellular $\Delta G_{ATP}$, low fractional OXPHOS was interpreted to reflect reduced bioenergetic efficiency in leukemia. Moreover, such findings indicate that traditional measurements of 'OXPHOS' capacity using intact cells (e.g. extracellular flux analysis) woefully underestimate true OXPHOS kinetics.

To differentiate between bioenergetic signatures inherent to proliferating cells and those which are unique to leukemia, experiments were repeated in mononuclear cells isolated from bone marrow aspirates collected from healthy volunteers. In these experiments, fractional OXPHOS in healthy bone marrow cells was comparable to PBMC and once again elevated above all leukemia cell lines (*Figure 2G*; '$BM_{Healthy}$', *Figure 2—figure supplement 1A–B*). As an additional control, identical experiments were carried out in primary human muscle precursor cells (human myoblasts – 'HMB' *Figure 2—figure supplement 1C–D*). These cells were cultured from muscle biopsies uniformly collected from the gastrocnemius muscle (10 cm distal to the tibial tuberosity) of healthy human subjects and were intended to serve as a non-cancerous, proliferative human progenitor control. Importantly, fractional OXPHOS was elevated above leukemia in human muscle progenitor cells (*Figure 2G*: 'HMB'), indicating that decreased bioenergetic efficiency is not an absolute requirement of cellular proliferation, but rather a unique bioenergetic feature of leukemic mitochondria.

## Exposure to physiological $\Delta G_{ATP}$ reveals direct inhibition of ETS flux by ATP in leukemic mitochondria

To gain insight into the mechanism of OXPHOS limitations in leukemia, at the end of the $\Delta G_{ATP}$ titration, oligomycin was added to inhibit ATP synthesis and maximal uncoupled respiration was stimulated with FCCP titration. Maximal FCCP-supported flux under these conditions is denoted as '$FCCP_{\Delta GATP}$' (*Figure 2D*). In the intact cell assay (*Figure 1C*), increased glycolytic flux is presumed to

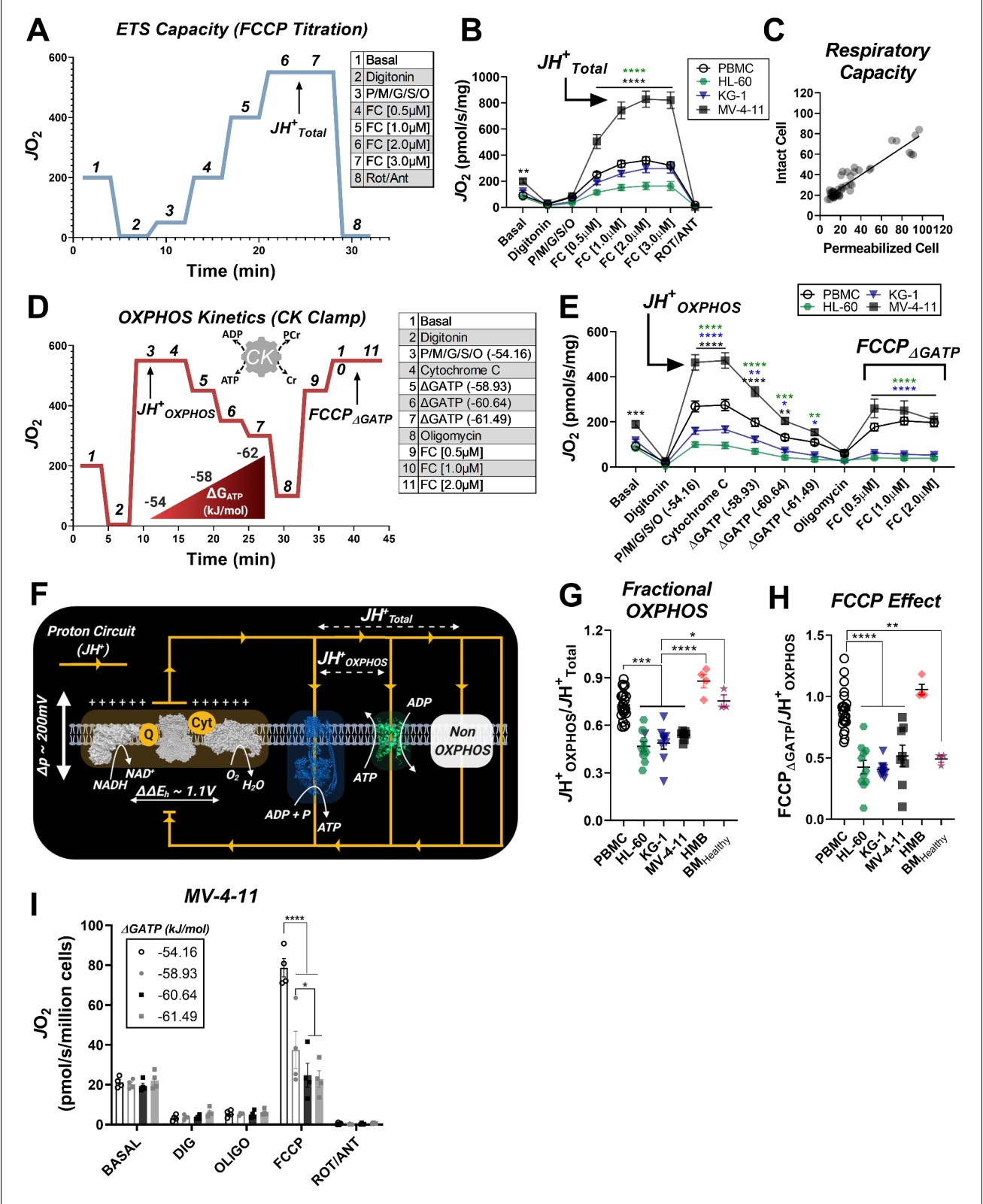

**Figure 2.** Impaired OXPHOS kinetics and ATP-dependent inhibition of ETS flux are unique phenotypes of leukemic mitochondria. All experiments were performed using digitonin-permeabilized cells. (A) Schematic depicting changes in oxygen consumption ($JO_2$) during an ETS capacity protocol (FCCP titration) where points 6–7 represent the maximum proton conductance of the respiratory system ($JH^+_{Total}$). (B) ETS capacity protocol measured in leukemia cell lines and PBMC. (C) Comparison of respiratory capacity between intact and permeabilized experimental conditions across cell types. (D)

*Figure 2 continued on next page*

*Figure 2 continued*

Schematic depicting $JO_2$ during an OXPHOS kinetics protocol ($\Delta G_{ATP}$ titration) where point three represents maximum proton conductance by the OXPHOS system ($JH^+_{OXPHOS}$) and point 10–11 represents maximum proton conductance of the respiratory system in the presence of $\Delta G_{ATP}$ (FCCP$_{\Delta GATP}$). (E) OXPHOS capacity protocol measured in leukemia cell lines and PBMC. (F) Illustration detailing maximal proton current generated by the electron transport system ($JH^+_{Total}$) and proportion of current harnessed by the phosphorylation system ($JH^+_{OXPHOS}$). (G) Comparison of fractional OXPHOS calculated as the ratio of $JH^+_{OXPHOS}$ to $JH^+_{Total}$. (H) Comparison of FCCP Effect calculated as the ratio of FCCP$_{\Delta GATP}$ to $JH^+_{OXPHOS}$. For all experiments, n = 10 for leukemia cell lines, n = 22 for PBMC, n = 4 for HMB, and n = 3 for BM$_{Healthy}$. (I) FCCP-stimulated flux was measured under four ATP-free energy ($\Delta G_{ATP}$) conditions in permeabilized MV-4–11 cells; n = 4 independent experiments. Data are presented as mean ± SEM and analyzed by two-way ANOVA in (B, E, I) and one-way ANOVA in (G-H). *$p < 0.05$, **$p < 0.01$, ***$p < 0.001$, ****$p < 0.0001$.

The online version of this article includes the following source data and figure supplement(s) for figure 2:

**Source data 1.** Raw values for '*Figure 2*' and '*Figure 2—figure supplement 1*'.
**Figure supplement 1.** Respiratory profile of healthy bone marrow mononuclear cells and primary human myoblasts.

maintain cellular $\Delta G_{ATP}$ during FCCP titration. Thus, the continued presence of extra-mitochondrial $\Delta G_{ATP}$ in our permeabilized cell system was intended to model the adenylate constraints present in intact cells. By comparing maximal OXPHOS flux ('$JH^+_{OXPHOS}$') to maximum FCCP-stimulated respiration in the presence of $\Delta G_{ATP}$ ('FCCP$_{\Delta GATP}$'), it becomes possible to quantitate any flux limitations imposed by physiological ATP/ADP. Importantly, ATP synthase is not functional during the assay, thus any flux limitations imposed by $\Delta G_{ATP}$ would be interpreted to reflect direct ETS regulation (e.g. allostery). In PBMC and human muscle progenitor cells, the addition of FCCP at the end of the $\Delta G_{ATP}$ titration restored respiration to levels obtained under low (−54.16 kJ/mol) ATP free energy (*Figure 2—figure supplement 1D–E*), indicating minimal ETS flux inhibition by $\Delta G_{ATP}$. Surprisingly, relative to $JH^+_{OXPHOS}$, FCCP-stimulated respiration was substantially blunted in the presence of high ATP-free energy across all three leukemia cell lines (*Figure 2E*), as well as healthy bone marrow cells (*Figure 2—figure supplement 1C*), resulting in a near twofold difference in the FCCP$_{\Delta GATP}$/$JH^+_{OXPHOS}$ ratio, termed 'FCCP Effect' throughout (*Figure 2H*). Importantly, in the absence of ATP, the addition of CK and PCR up to 21 mM did not impact FCCP-supported flux in permeabilized MV-4–11 cells (*Figure 2—figure supplement 1E*). To determine the sensitivity of ETS inhibition by $\Delta G_{ATP}$, FCCP-supported flux in MV-4–11 cells was assessed at defined ATP-free energies. In these experiments, extramitochondrial $\Delta G_{ATP}$ was administered after CV inhibition with oligomycin, followed by FCCP titration. Results revealed a dose-dependent decrease in uncoupled respiration in response to increasing $\Delta G_{ATP}$ (*Figure 2I*), confirming that ATP-free energy is both necessary and sufficient to induce direct ETS flux inhibition in leukemia.

Inhibition of respiratory flux mediated by $\Delta G_{ATP}$ could reflect a number of potential mechanisms ranging from cytoskeletal alterations, direct inhibition of the matrix dehydrogenase network (e.g. inhibitory phosphorylation of pyruvate dehydrogenase), and/or ETS inhibition (*Fisher-Wellman et al., 2018*). To differentiate between these potential outcomes, mitochondria were isolated from PBMC and each of the three leukemia cell lines and similarly assessed for OXPHOS kinetics. In mitochondria energized with saturating carbon, increasing $\Delta G_{ATP}$ led to a more pronounced decrease in respiration in mitochondria of all three leukemia cell lines (*Figure 3A*). The ability of FCCP to restore maximal respiratory flux was also once again blunted in leukemic mitochondria (*Figure 3A–B*), consistent with lower fractional OXPHOS (*Figure 2G*). Evidence of ATP-dependent inhibition of ETS flux using both isolated mitochondria and permeabilized cells was used as criteria to rule out any involvement of the cytoskeleton. To differentiate between respiratory flux inhibition localized to the matrix dehydrogenases or the ETS, NADH/NAD$^+$ redox poise was measured in substrate-replete isolated mitochondria exposed to an identical $\Delta G_{ATP}$ span. Results are depicted as a percentage of complete reduction, where 0% reduction reflects isolated mitochondria at 37°C without added substrates and 100% reduction is recorded at the end of the assay with the addition of the CIV inhibitor cyanide. Except for a slight hyper-reduction in HL-60 mitochondria, NADH/NAD$^+$ redox was similar across groups (*Figure 3C*), indicating that ATP-mediated respiratory flux inhibition in leukemia is not due to a generalized impairment in carbon substrate uptake and/or dehydrogenase flux. Having eliminated the cytoskeleton and the dehydrogenase network as potential sites of inhibition, we next turned our attention to the ETS. To determine if flux inhibition induced by $\Delta G_{ATP}$ was specific to a given respiratory complex, OXPHOS kinetics were assessed in isolated mitochondria energized with either complex I (CI)- or CII-linked substrate combinations. Note, the presence

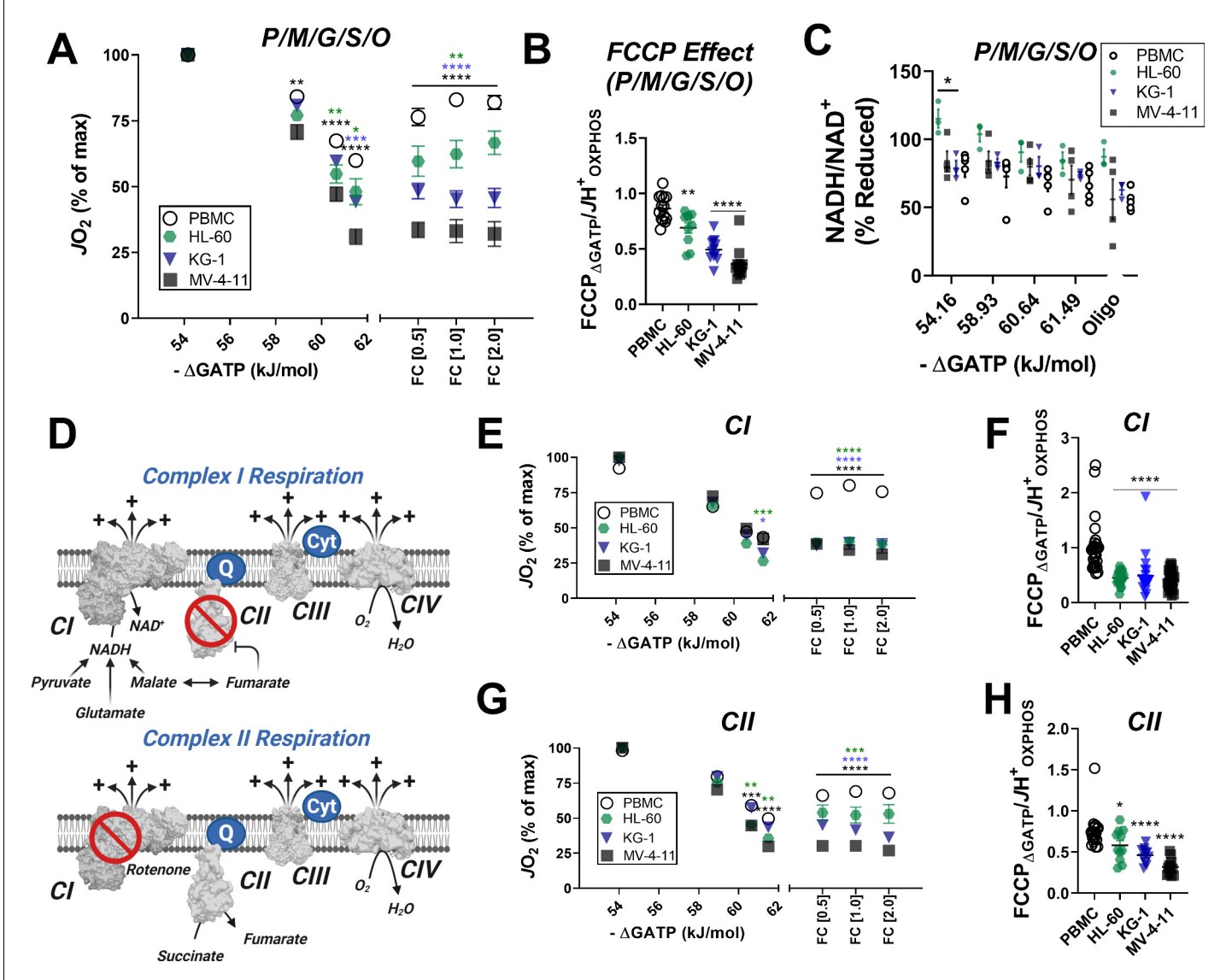

**Figure 3.** In leukemic mitochondria $\Delta G_{ATP}$ regulates ETS flux independent of substrate condition. (**A**) OXPHOS kinetics supported by P/M/G/S/O in mitochondria isolated from PBMC and leukemia cells. (**B**) Comparison of FCCP Effect calculated as the ratio of FCCP $_{\Delta GATP}$ to $JH^+_{OXPHOS}$ from B. (**C**) Relationship between $\Delta G_{ATP}$ and NADH/NAD + redox supported by P/M/G/S/O in mitochondria isolated from leukemia cell lines and PBMC. (**D**) Schematic depicting CI supported respiration driven by pyruvate/glutamate/malate and inhibition of CII by equilibration of malate/fumarate (top) and CII supported respiration driven by succinate and inhibition of CI by rotenone (bottom). OXPHOS kinetics of mitochondria isolated from PBMC and leukemia cells and supported by Complex I substrates (**E**) and Complex II substrates (**G**). FCCP Effect of complex I (**F**) and complex II (**H**) supported respiration. (**A, B, E–H**) n = 7–10 for leukemia cell lines and n = 22 for PBMC. (**C**) n = 3–5 independent experiments. Data are mean ± SEM and analyzed by one-way ANOVA in (**A**) and two-way ANOVA in (**B–C, E–H**). *p<0.05, **p<0.01, ***p<0.001, ****p<0.0001.

The online version of this article includes the following source data for figure 3:

**Source data 1.** Raw values for '*Figure 3*'.

of saturating malate in the CI substrate mix results in CII inhibition via malate-fumarate equilibration (*Figure 3D*). Likewise, the addition of rotenone in the presence of succinate eliminates residual CI-supported flux by downstream products of succinate oxidation (*Figure 3D*). Using either CI- or CII-linked substrates, we once again observed a more pronounced decrease in respiration in response to $\Delta G_{ATP}$ titration in leukemic mitochondria, as well as a relative inability of FCCP to restore maximal respiratory flux (*Figure 3E–H*). Taken together these findings demonstrate that leukemic

mitochondria are characterized by a unique form of OXPHOS regulation involving ETS inhibition by ATP free energy.

## Subcellular proteomics reveals unique isoform expression of the adenine nucleotide translocase (ANT) in leukemia

To identify potential protein mediators responsible for OXPHOS insufficiency in leukemia, we conducted a proteomics screen using TMT-labeled peptides prepared from the same isolated mitochondria samples used for functional characterization. To control for group differences in percent mitochondrial enrichment, nLC-MS/MS raw data were searched using the MitoCarta 2.0 database, as previously described (*McLaughlin et al., 2020*). Using this approach, total mitochondrial protein abundance was similar between groups (*Supplementary file 1*), thus allowing for intrinsic mitochondrial signatures to be identified across leukemia. In total, 135 differentially expressed mitochondrial proteins (adjusted p value < 0.01) were identified comparing PBMC to each of the three leukemia lines (*Figure 4A*). For pair-wise comparisons of mitochondrial protein expression between PBMC and each of the three leukemia cell lines, see *Supplementary file 1*. With respect to the shared

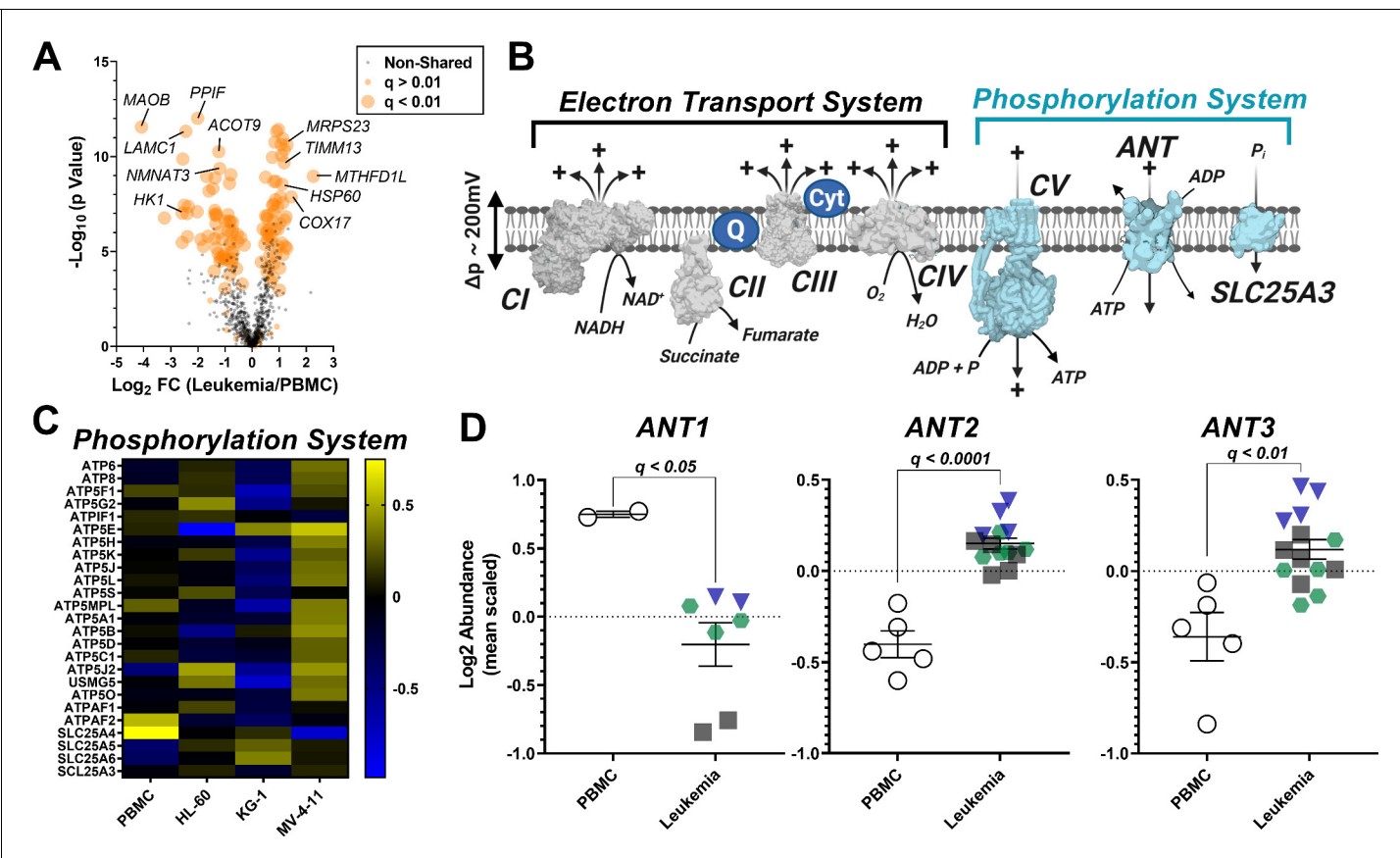

**Figure 4.** Analysis of mitochondrial proteome reveal disparate expression of ANT isoforms in leukemia. TMT-labeled nLC-MS/MS was performed on mitochondrial lysates from each cell type. (A) Volcano plot depicting changes in proteome between leukemia cell lines and PBMC with mitochondrial proteins shown in orange. Significance is indicated by size of each circle with changes in significance (p<0.01) represented by larger circles. (B) Schematic depicting the OXPHOS system with enzymes integral to the ETS shown in gray and the phosphorylation system shown in blue. (C) Heat map displaying the common differentially expressed proteins across the phosphorylation system of leukemia and PBMC. Data are displayed as Log$_2$ protein intensity of all quantified master proteins. (D) Comparison of log2 abundance of ANT isoforms in leukemia and PBMC; n = 4–6 mitochondrial preparations per cell lines. Data are presented as mean ± SEM and analyzed by unpaired t-tests with multiple hypothesis correction (P$_{adjusted}$, Benjamini Hochberg FDR correction, significance cutoff of q < 0.1).

The online version of this article includes the following source data and figure supplement(s) for figure 4:

**Source data 1.** Raw values for '*Figure 4*' and '*Figure 4—figure supplement 1*'.

**Figure supplement 1.** Heatmap analysis depicting abundance of subunits and assembly factors that comprise the ETS complexes.

differentially expressed proteins, several of these proteins have previously been implicated in cancer biology, such as decreased MAOB (*Ryu et al., 2018*) and HK1 (*Rai et al., 2019*), and increased MTHFD1L (*Lee et al., 2017*), and COX17 (*Singh et al., 2020*; *Figure 4A*).

Focusing on the OXPHOS proteome, we assessed the abundance of the individual protein subunits that comprise CI, CII, CIII, and CIV, as well as the protein components of the phosphorylation system which include ATP synthase (CV), the phosphate carrier (SLC25A3), and ANT (*Figure 4B*). Although considerable heterogeneity was present across groups, comparing protein expression profiles of the individual subunits that comprise CI-CV and SLC25A3 (*Figure 4C*, *Figure 4—figure supplement 1A–D*) revealed that only 6 of the 110 subunits were similarly altered in leukemia (*Supplementary file 1*). With the exception of COX6A1, all protein subunits were involved in the assembly of CI (NDUFB10), CIV (COA4, COA7, COX17) or CV (ATPAF2). In stark contrast, the expression profiles of the three main ANT isoforms were entirely distinct between PBMC and leukemia mitochondria, highlighted by reduced ANT1 (SLC25A4) and increased ANT2 (SLC25A5) and ANT3 (SLC25A6) in leukemia (*Figure 4D*).

## Inhibition of ETS flux by $\Delta G_{ATP}$ is a result of matrix ATP consumption in leukemia

Given that ATP-free energy was required to induce ETS flux inhibition, we hypothesized that this effect may be mediated by ATP transport into the matrix, facilitated by dominant ANT2/3 expression in leukemia (*Chevrollier et al., 2011*). To test this hypothesis, FCCP-supported respiration was assessed in permeabilized MV-4–11 and HL-60 cells exposed to $\Delta G_{ATP}$ of −61.49 kJ/mol in the absence and presence of the ANT inhibitor carboxyatractyloside (CAT) (*Maldonado et al., 2016*). Consistent with our prior findings, the addition of FCCP in the presence of $\Delta G_{ATP}$ was incapable of restoring flux to levels obtained with minimal ATP-free energy in leukemia (*Figure 5A–B*; '$\Delta G_{ATP}$ (−61.49 kJ/mol)'). However, relative to no adenylates, as well as minimal $\Delta G_{ATP}$ (e.g. - 54.16 kJ/mol), the addition of CAT restored maximal FCCP-supported ETS flux in the presence of maximal ATP free energy (*Figure 5A–C*). Similar experiments performed in MV-4–11 isolated mitochondria (*Figure 5D*), as well as with the ANT inhibitor bongkrekic acid (*Figure 5—figure supplement 1A*) revealed nearly identical results, indicating that ETS flux inhibition by $\Delta G_{ATP}$ requires matrix ATP uptake via ANT.

As a component of OXPHOS, ANT functions to exchange matrix ATP for cytosolic ADP in a process that consumes (i.e. depolarizes) the electrochemical proton gradient across the inner membrane. It is critical to point out that the directionality of transport by ANT is dependent on inner membrane polarization. Based on this, if indeed ANT2/3 were favoring matrix ATP uptake in leukemia then chemical inhibition of the phosphorylation system should depolarize, rather than hyperpolarize, membrane potential in leukemic isolated mitochondria. To test this, we quantified mitochondrial membrane potential in substrate-replete isolated mitochondria from HL-60 and KG-1, as well as PBMC, across a $\Delta G_{ATP}$ span and then assessed the impact of ANT inhibition with CAT or CV inhibition with oligomycin. Contrary to that seen in PBMC (*Figure 5E*), the addition of CAT or oligomycin caused a near complete elimination of mitochondrial membrane potential in HL-60 and KG-1 (*Figure 5F–G*, *Figure 5—figure supplement 1B*), indicative of matrix ATP uptake/consumption. Interestingly, in similar experiments with MV-4–11 mitochondria, the addition of oligomycin partially polarized membrane potential (*Figure 5H*), presumably due to the higher ETS capacity of MV-4–11 compared to HL-60 or KG-1 (*Figure 2B*). However, in the absence of any carbon substrates (i.e. no forward electron transport), membrane potential generated exclusively by $\Delta G_{ATP}$ was ~2 fold more polarized in all AML lines compared to PBMC (*Figure 5I*). In fact, for each AML line, membrane potential generated by ATP consumption was comparable to that generated during forward electron transport with saturating carbon substrates (*Figure 5J*). Together, these data indicate that leukemic mitochondria are particularly poised to consume, rather than produce, ATP. Such a mechanism appears mediated by the dominant expression of ANT2/3 in AML and once inside the matrix, ATP exerts a direct inhibitory effect on ETS flux.

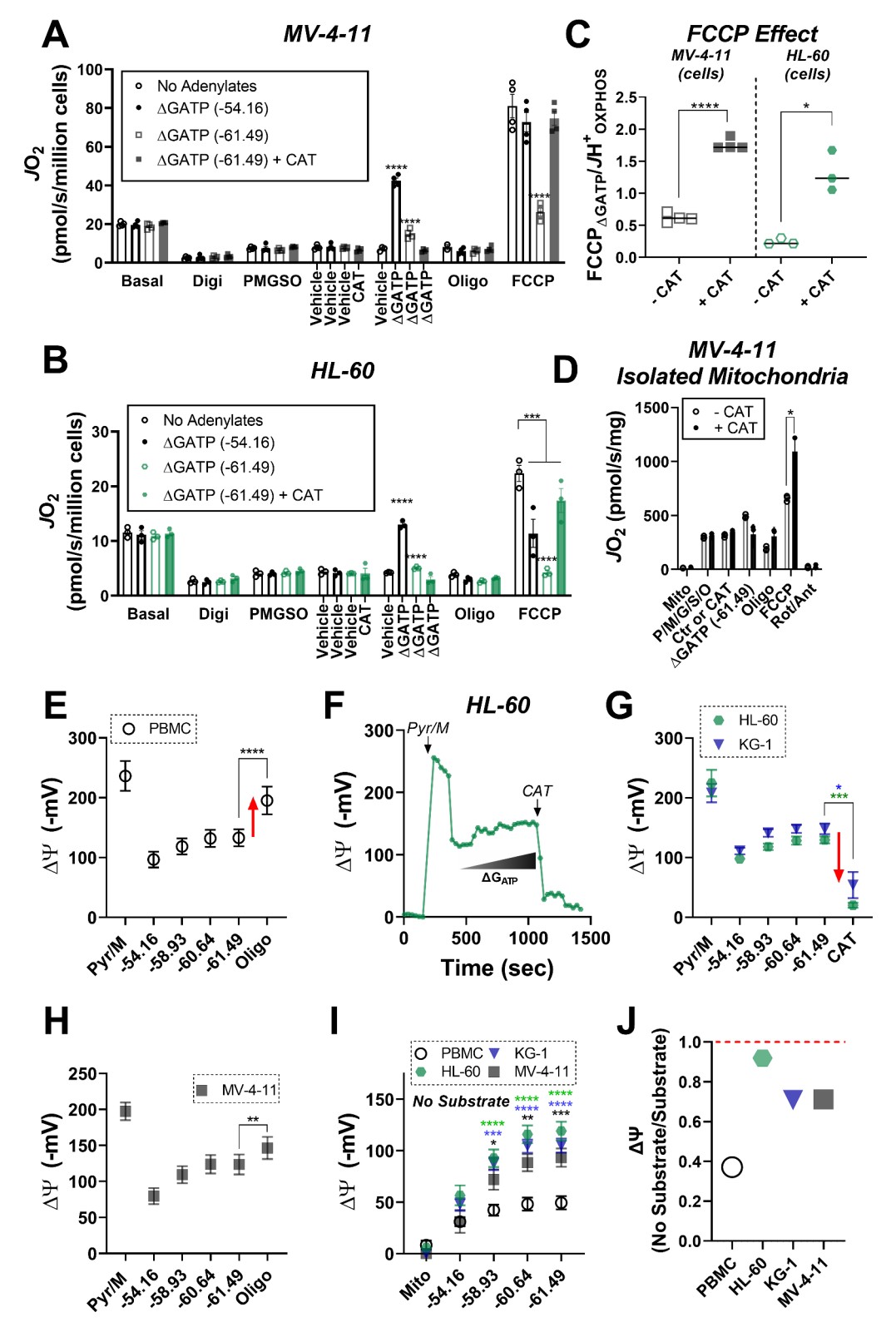

**Figure 5.** ETS flux inhibition by $\Delta G_{ATP}$ links to matrix ATP consumption in leukemia. (A–B) OXPHOS kinetics (via CK clamp) were assessed in the absence of adenylates or in the presence of minimal $\Delta G_{ATP}$ (−54.16), maximal $\Delta G_{ATP}$ (−61.49), or maximal $\Delta G_{ATP}$ + CAT (Carboxyatractyloside; ANT inhibitor). Comparison of OXPHOS kinetics in MV-4–11 (A) and HL-60 cells (B). (C) Ratio of FCCP $_{\Delta GATP}$ to $JH^{+}_{OXPHOS}$ with and without CAT in MV-4–11 and HL-60 cells. (D) OXPHOS kinetics measured in the presence of CAT in mitochondria isolated from MV-4–11. (A–D) n = 4 independent experiments

*Figure 5 continued on next page*

Figure 5 continued

per cell type. (E) Mitochondrial membrane potential ($\Delta\Psi$) in PBMC isolated mitochondria across a $\Delta G_{ATP}$ span, followed by CV inhibition with oligomycin; n = 8 independent experiments. (F) Representative trace of $\Delta\Psi$ in HL-60 isolated mitochondria across a $\Delta G_{ATP}$ span, followed by ANT inhibition with CAT. (G) Quantification of the experiment depicted in panel F in HL-60 and KG-1 isolated mitochondria; n = 3 independent experiments. (H) Mitochondrial $\Delta\Psi$ in MV411 isolated mitochondria across a $\Delta G_{ATP}$ span, followed by CV inhibition with oligomycin; n = 9 independent experiments. (I) Mitochondrial $\Delta\Psi$ in isolated mitochondria across a $\Delta G_{ATP}$ span in the absence of any carbon substrates; n = 4 independent experiments. (J) Ratio of group mean $\Delta\Psi$ generated at $\Delta G_{ATP} = -61.49$ kJ/mol in the absence versus presence of carbon substrates (pyruvate, malate). Data are presented as mean ± SEM and analyzed by two-way ANOVA (A–B, I) or paired t-tests (C–E, G–H). *p<0.05, **p<0.01, ***p<0.001, ****p<0.0001.

The online version of this article includes the following source data and figure supplement(s) for figure 5:

**Source data 1.** Raw values for '*Figure 5*' and '*Figure 5—figure supplement 1*'.
**Figure supplement 1.** ETS flux in the presence of $\Delta G_{ATP}$ is restored by ANT inhibition ANT.

## Low fractional OXPHOS in AML is reversed by small-molecule inhibitors of TRAP1

Having established that extramitochondrial $\Delta G_{ATP}$ must gain access to the matrix space to inhibit ETS flux in leukemia, we next set out to elucidate the potential protein mediator(s) of this effect. To do this, we searched our proteomics dataset for mitochondrial proteins with known kinase and/or ATPase function that were substantially upregulated across all three leukemia lines and identified mitochondrial TRAP1 (*Figure 6A*). TRAP1 is the mitochondrial paralog of the heat shock protein 90 (HSP90) family and is widely recognized as a potential anticancer drug target across multiple human malignancies, including leukemia (*Bryant et al., 2017*; *Kim et al., 2020*; *Li et al., 2020*; *Sanchez-Martin et al., 2020a*; *Sanchez-Martin et al., 2020b*; *Sciacovelli et al., 2013*; *Yoshida et al., 2013*). Given that ATPase activity is required for TRAP1 function (*Ramkumar et al., 2020*), we hypothesized that $\Delta G_{ATP}$-mediated ETS inhibition may be driven by acute activation of TRAP1. To test this hypothesis, OXPHOS kinetics were assessed in permeabilized MV-4–11 cells in the absence and presence of the purported TRAP1 inhibitor 17-AAG (*Kamal et al., 2003*). In substrate-replete permeabilized MV-4–11 cells, acute exposure to 17-AAG had no impact on maximal FCCP-supported respiration in the absence of adenylates (*Figure 6—figure supplement 1A*). Remarkably, the presence of 17-AAG increased $JH^+_{OXPHOS}$ and calculated fractional OXPHOS relative to vehicle control and completely restored FCCP-supported respiration in the presence of ATP-free energy (*Figure 6B–D*). Similar results were observed using permeabilized HL-60 cells (*Figure 6—figure supplement 1B–D*), as well as using the mitochondrial-targeted TRAP1 inhibitor gamitrinib (*Figure 5—figure supplement 1A*). Using both MV-4–11 and KG-1 permeabilized cells, the ability of 17-AAG to restore OXPHOS flux was similar in the presence of CI or CII-based carbon substrates (*Figure 6—figure supplement 1E–F*).

Although ATP is widely understood to be the universal energy currency in cells, it is critical to consider that ATP alone has minimal bio-synthetic power; rather, its utilization as a common energy currency is solely a function of the remarkable displacement of the molecule from equilibrium (~10 orders of magnitude) (*Willis et al., 2016*). This means that biological processes driven by ATP hydrolysis are presumably fueled by ATP-free energy, rather than ATP levels per se. The primary advantage of the CK clamp technique is that it allows for mitochondrial bioenergetics to be evaluated across a physiological $\Delta G_{ATP}$ span without appreciable changes in ATP concentration (*Figure 6—figure supplement 1G*). Thus, we reasoned that the CK clamp could be utilized to assess 17-AAG effectiveness across a $\Delta G_{ATP}$ span under conditions in which free [ATP] is not rate-limiting. For contrast, we compared the respiratory impact of 17-AAG to that of the commonly used ETC inhibitor antimycin A. Although antimycin A decreased respiratory flux in permeabilized MV-4–11 cells, percent inhibition by the compound was largely insensitive to $\Delta G_{ATP}$, consistent with its known mechanism of action at CIII (*Figure 6—figure supplement 1H*). In contrast, across all leukemia cell lines, restoration of respiratory flux by 17-AAG was exquisitely sensitive to $\Delta G_{ATP}$ (*Figure 6E*, *Figure 6—figure supplement 1I–J*).

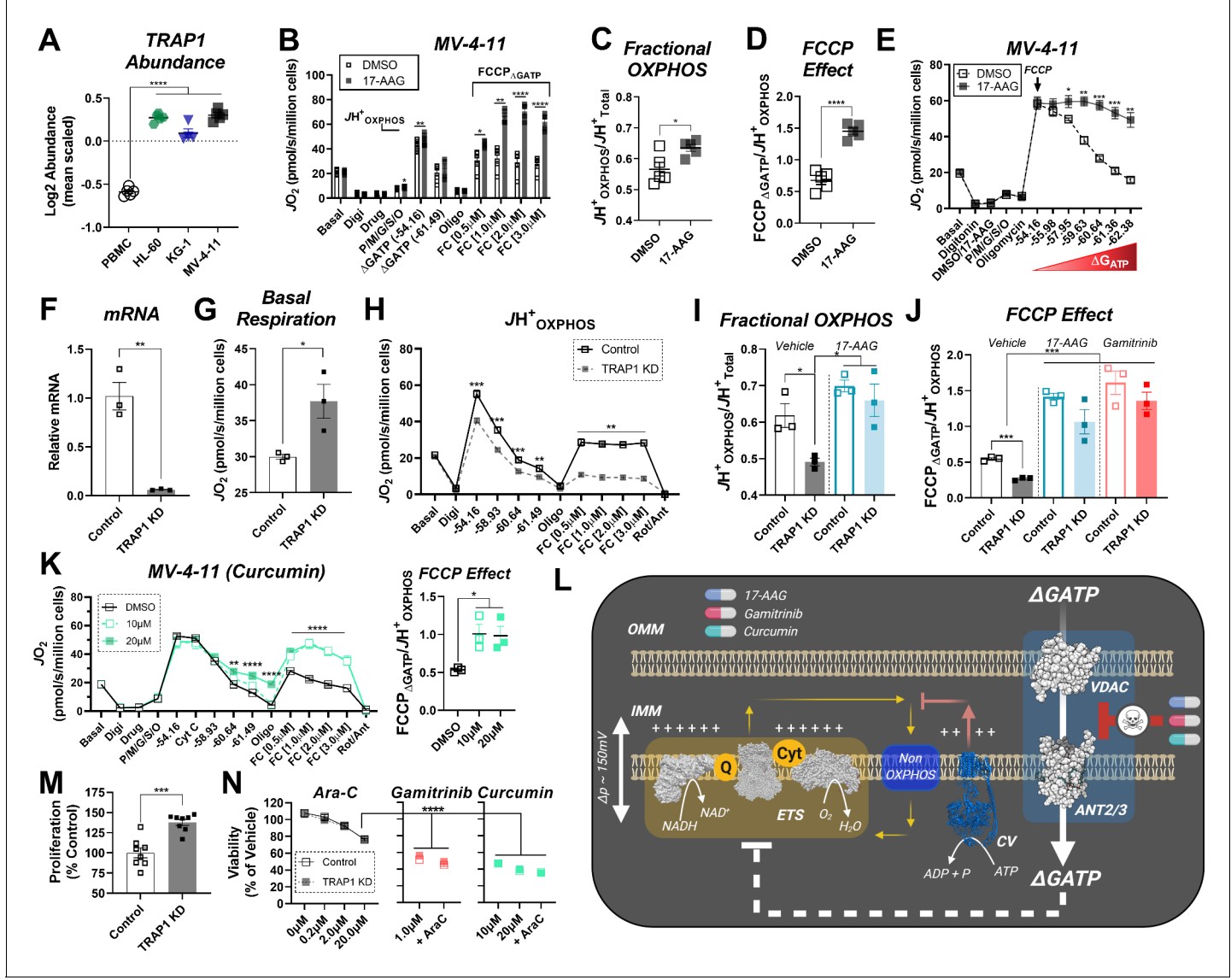

**Figure 6.** 17-AAG and gamitrinib increase fractional OXPHOS and restore ETS flux in the presence of $\Delta G_{ATP}$. (A) Log2 Abundance of TRAP1 in PBMC and leukemia cells. (B) Comparison of OXPHOS kinetics in presence of the TRAP1 inhibitor, 17-AAG (15 µM); n = 4 independent cell experiments. Comparison of (C) Fractional OXPHOS and (D) FCCP Effect in MV-4–11 cells treated with DMSO or 17-AAG (15 µM). (E) Comparison of respiratory flux inhibition within MV-4–11 cells across a range of $\Delta G_{ATP}$ in the absence and presence of 17-AAG (15 µM). Respiration was stimulated by the addition of FCCP (1 µM), followed by PCR titration to manipulate $\Delta G_{ATP}$. (F) Relative abundance of TRAP1 mRNA following shRNA knockdown of TRAP1 in MV-4–11 cells. (G) Basal respiration in intact cells. (H) OXPHOS kinetics in permeabilized MV-4–11 cells infected with lentivirus encoding shRNA targeted to TRAP1 (TRAP1 KD) or scrambled shRNA (Control). (I–J) Fractional OXPHOS and FCCP Effect measured in the presence of vehicle (DMSO), 17-AAG (15 µM), or gamitrinib (1 µM) in control and TRAP1 KD cells. (K) OXPHOS kinetics in permeabilized MV-4–11 cells in the presence of DMSO or curcumin (10–20 µM). FCCP Effect is graphed to the right. (L) Schematic depicting the presumed mechanism of action of 17-AAG, gamitrinib and curcumin in which the compounds selectively block ATP uptake via the VDAC-ANT axis to restore OXPHOS kinetics. (M) Cell proliferation expressed as a percentage of Control. (N) Cell viability in MV-4–11 cells infected with lentivirus encoding shRNA targeted to TRAP1 or scrambled shRNA and treated for 24 hr with increasing concentrations of Ara-C. Additional treatments included gamitrinib (1 µM), 17-AAG (15 µM) or curcumin (10–20 µM) either alone or plus Ara-C (20 µM). Data depicted as viability based on the percentage of vehicle using the propidium iodide assay. (A) n = 4/6/group; (B–D) n = 4 independent experiments; (E–K) n = 3 independent experiments; (M) n = 8 independent experiments; (N) n = 4–6 independent experiments. Data are presented as mean ± SEM analyzed by unpaired t-tests (F–H, M) two-way ANOVA (B, K), one-way ANOVA (A, I, J, N), paired t-tests (B–E). *$p<0.05$, **$p<0.01$, ***$p<0.001$, ****$p<0.0001$.

The online version of this article includes the following source data and figure supplement(s) for figure 6:

**Source data 1.** Raw values for '*Figure 6*' and '*Figure 6—figure supplement 1*'.

**Figure supplement 1.** Effects of 17-AAG on OXPHOS kinetics.

## OXPHOS restoration by 17-AAG and gamitrinib is independent of TRAP1

To control for potential off-target effects mediated by 17-AAG or gamitrinib, MV-4–11 cells were infected with lentivirus containing pooled short hairpin RNA (shRNA) against TRAP1. Control cells were infected with lentivirus containing scrambled shRNA. All constructs encoded GFP, as well as a puromycin selection gene to allow for stable selection. Following 24 hr exposure to lentiviral particles and multiple rounds of puromycin selection, shRNA against TRAP1 led to a > 90% reduction in TRAP1 mRNA (*Figure 6F*). Consistent with prior reports (*Laquatra et al., 2021*; *Sciacovelli et al., 2013*), TRAP1 knockdown increased basal respiration in intact cells (*Figure 6G*). To determine the impact of TRAP1 knockdown on OXPHOS kinetics, $JH^+_{Total}$ and $JH^+_{OXPHOS}$ were assessed in substrate-replete permeabilized cells. Despite no change in $JH^+_{Total}$ (*Figure 6—figure supplement 1K*), TRAP1 knockdown decreased $JH^+_{OXPHOS}$ and exacerbated the ability of ATP-free energy to blunt ETS flux (*Figure 6H–J*). This was surprising, given that acute administration of 17-AAG/gamitrinib consistently increased fractional OXPHOS across all AML lines (*Figure 6C*, *Figure 6—figure supplement 1C*). Based on this, we hypothesized that the ability of 17-AAG and/or gamitrinib to bolster OXPHOS kinetics may be independent of TRAP1. To test this, we repeated the $JH^+_{OXPHOS}$ experiments in control and TRAP1 knockdown cells in the presence of either 17-AAG or gamitrinib. In control cells, acute administration of 17-AAG and gamitrinib once again increased fractional OXPHOS and restored the FCCP effect and near identical results were also apparent in TRAP1 knockdown cells (*Figure 6I–J*, *Figure 6—figure supplement 1L*). Such findings indicated that OXPHOS restoration by 17-AAG/gamitrinib occurs independent of TRAP1 and is thus attributable to an 'off-target' mechanism. One such off-target effect documented for 17-AAG relates to the ability of the compound to bind voltage-dependent anion transporter (VDAC) (*Xie et al., 2011*), the principal gatekeeper to adenylate transport across the outer mitochondrial membrane. A similar mechanism has also been described for the anticancer compound curcumin (*Tewari et al., 2015*); however, the implications of these interactions on cellular bioenergetics remain incompletely understood. Interestingly, nearly identical to the effect observed in the presence of 17-AAG/gamitrinib, acute exposure to curcumin also reversed $\Delta G_{ATP}$-mediated ETS inhibition in MV-4–11 cells (*Figure 6K*). Based on the striking functional similarities between 17-AAG, gamitrinib and curcumin, such findings suggest a potential novel mechanism whereby these small molecules are uniquely capable of selectively blocking matrix ATP uptake, while remaining permissive to ADP uptake, presumably via interaction with the VDAC-ANT axis (*Figure 6L*). In support of this, curcumin-VDAC interaction is known to lock VDAC in the 'closed' confirmation, in turn making it partially selective for cation (i.e. $ADP^{3-}$), rather than anion (i.e. $ATP^{4-}$), transport (*Tewari et al., 2015*).

To investigate the relationship between mitochondrial fractional OXPHOS and AML biology, we assessed cellular proliferation and viability in control and TRAP1 knockdown cells exposed to gamitrinib, curcumin, or vehicle control. In the absence of any small-molecule intervention, cellular proliferation was higher in TRAP1 knockdown cells (*Figure 6M*), consistent with low fractional OXPHOS being advantageous to AML growth. To determine the therapeutic efficacy of restoring fractional OXPHOS in AML, cells were exposed for 24 hr to either gamitrinib or curcumin, either alone or in combination with the front-line AML chemotherapeutic cytarabine (Ara-C). Compared to marginal cytotoxicity in response to Ara-C dose escalation, 24 hr exposure to either gamitrinib or curcumin decreased cell viability by as much as 60% (*Figure 6N*). Note, identical cytotoxicity was observed in control and TRAP1 knockdown cells, and the additional presence of Ara-C had no further impact (*Figure 6N*).

## Intrinsic limitations in OXPHOS characterize human primary leukemia

To determine if the bioenergetic phenotypes present in leukemia cell lines translated to the clinic, we recruited patients diagnosed with leukemia and comprehensively evaluated mitochondrial bioenergetic function in mononuclear cells isolated from bone marrow aspirates. All patients had confirmed leukemia at the time of sample acquisition. Biochemical results were compared to PBMC isolated from age-matched participants without a prior history of leukemia. As additional controls, OXPHOS kinetics were assessed in mononuclear cells isolated from bone marrow aspirates of heathy subjects ($BM_{Healthy}$), as well as purified CD34 +hematopoietic stem cells. To ascertain the impact of clonal cell expansion on the underlying mitochondrial network, experiments in unstimulated (i.e.

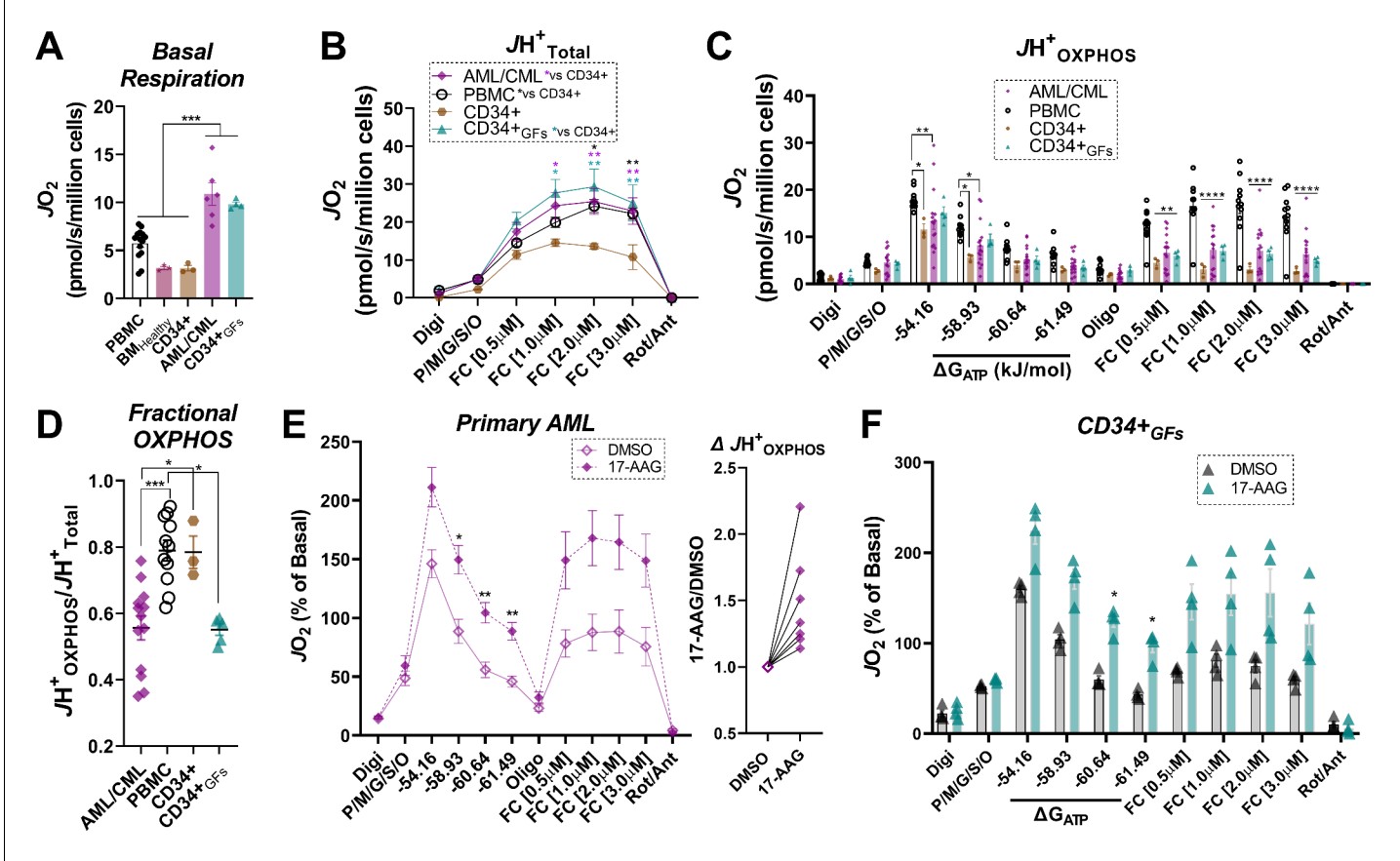

**Figure 7.** Human primary leukemia is characterized by low Fractional OXPHOS. (A) Basal respiration in intact cells – 'PBMC' (age-matched to the primary leukemia samples); 'BM_Healthy' (bone marrow mononuclear cells); 'CD34+' (pure CD34+ cells not exposed to growth factors); 'P. Leukemia' (mononuclear cells isolated from bone marrow of patients with confirmed leukemia); 'CD34+GFs' (pure CD34+ cells stimulated to undergo clonal expansion in culture). (B) ETS capacity assay in permeabilized cells. (C) OXPHOS kinetics in permeabilized cells. (D) Fractional OXPHOS. (A–D) n = 3–15 independent experiments. (E) Comparison of OXPHOS kinetics in the presence of DMSO or 17-AAG (15 µM) in a cohort of patients with confirmed AML. Graph to the right depicts $JH^+_{OXPHOS}$ depicted as a fold change from DMSO; n = 7 independent cell experiments. (F) Comparison of OXPHOS kinetics in the presence of DMSO or 17-AAG (15 µM) in CD34+GFs; n = 4 independent cell experiments. (E–F) Data depicted as a percentage of basal respiration based on oxygen consumption rates obtained in intact cells. Data are presented as mean ± SEM analyzed by one-way ANOVA (A, D) two-way ANOVA (B–C), or paired t-tests (E–F). *p<0.05, **p<0.01, ***p<0.001, ****p<0.0001.

The online version of this article includes the following source data and figure supplement(s) for figure 7:

**Source data 1.** Raw values for '*Figure 7*' and '*Figure 7—figure supplement 1*'.

**Figure supplement 1.** Effects of 17-AAG on OXPHOS kinetics and cell viability in PBMC and healthy bone marrow mononuclear cells.

quiescent) CD34+ cells were also performed following chronic exposure to proliferative stimuli (e.g. TPO, SCF, FLT-3; 'CD34 +GFs'). Relative to PBMC, BM_Healthy, and quiescent CD34+, intact basal respiration was elevated in primary leukemia, as well as CD34+GFs (*Figure 7A*), reminiscent of that seen in AML cell lines and entirely consistent with their proliferative phenotypes. Relative to quiescent CD34+, $JH^+_{Total}$ in substrate-replete permeabilized cells was ~2 fold higher in PBMC, CD34+GFs and primary leukemia (*Figure 7B*), consistent with higher cellular mitochondrial density. Despite this apparent increase in mitochondrial content, $JH^+_{OXPHOS}$ remained elevated above quiescent CD34 +only in PBMC, as OXPHOS kinetics were completely unaltered comparing CD34+GFs or primary leukemia to quiescent CD34+ (*Figure 7C*). As seen in the AML cell lines, calculated fractional OXPHOS was decreased in CD34+GFs and primary leukemia, entirely consistent with low fractional

OXPHOS being required for hematopoietic clonal expansion (*Figure 7D*). To determine the impact of 17-AAG on OXPHOS kinetics, CK clamp experiments were performed in a separate cohort of primary AML samples, as well as clonally expanding CD34+ cells. Relative to vehicle control, 17-AAG increased $JH^+_{OXPHOS}$ and restored FCCP-stimulated respiration in the presence of $\Delta G_{ATP}$ in primary AML and CD34+$_{GFs}$ (*Figure 7E–F*). Of note, acute 17-AAG had no impact on OXPHOS kinetics in PBMC or BM$_{Healthy}$ and 24 hr exposure of PBMC to 17-AAG or gamitrinib did not impact cell viability (*Figure 7—figure supplement 1A–C*).

## Intrinsic limitations in OXPHOS power output characterize venetoclax resistance in AML

The preponderance of evidence related to mitochondrial bioenergetics in AML indicates that active repression of OXPHOS is advantageous to the leukemia phenotype. Related to this, although several reports have linked AML chemoresistance to apparent increases in 'OXPHOS reliance' (*Farge et al., 2017*; *Guièze et al., 2019*; *Liu et al., 2020*; *Roca-Portoles et al., 2020*), such conclusions are based largely on intact cellular respirometry in which OXPHOS kinetics are not directly quantified. To address the specific role for OXPHOS in AML chemoresistance, we conducted a series of experiments in chemosensitive HL-60 cells and compared the results to HL-60 cells made to be refractory to venetoclax (*Figure 8A–B*). In these experiments, $JH^+_{Total}$ and $JH^+_{OXPHOS}$ were determined as described above. In addition, OXPHOS kinetic data generated using the CK clamp was integrated with parallel analysis of OXPHOS efficiency (mitochondrial ATP synthesis relative to oxygen consumption; quantified as the P/O ratio). By incorporating empirically derived P/O, respiratory flux at each $\Delta G_{ATP}$ can be converted to ATP production rate. Assuming extra-mitochondrial force applied via the CK clamp is fixed at each titration step, ATP production rate can be used to quantitate OXPHOS power output in Watts ($J \cdot s^{-1}$). Power is an extremely useful parameter because it encompasses thermodynamic, kinetic, as well as stochiometric descriptions that effectively report on the actual quantities of OXPHOS work performed across the demand range. Looking first at intact cellular respirometry, both basal and maximal respiratory capacity were elevated in venetoclax-resistant HL-60 (HL60$_{VR}$) (*Figure 8C*). Remarkably, despite a near ~2 fold increase in cellular respiratory capacity, OXPHOS kinetics were in fact decreased in HL60$_{VR}$, such that $JH^+_{OXPHOS}$ failed to reach the respiratory rates observed under basal conditions prior to digitonin permeabilization (*Figure 8D*). Of note, the addition of extramitochondrial cytochrome C did not impact $JH^+_{OXPHOS}$ in either cell type (*Figure 8E*). The combination of increased maximal respiratory capacity combined with decreased $JH^+_{OXPHOS}$ translated to a striking decrease in both fractional OXPHOS (*Figure 8F*), as well as OXPHOS power output in HL60$_{VR}$ (*Figure 8G–I*), indicating that intrinsic OXPHOS insufficiency also underlies cellular bioenergetics in chemoresistant AML.

## Discussion

Increased mitochondrial oxidative metabolism, an established metabolic hallmark of leukemia (*Byrd et al., 2013*; *Kuntz et al., 2017*; *Lee et al., 2015*; *Sriskanthadevan et al., 2015*; *Suganuma et al., 2010*), has been historically interpreted to reflect an increased reliance on mitochondrial ATP production. However, fractional OXPHOS kinetics had not been empirically evaluated in leukemia at the onset of this project. Thus, it remained to be determined whether higher basal respiration in leukemia reflected accelerated demand for ATP regeneration or intrinsic OXPHOS insufficiency. Both conditions would be expected to similarly restrict cellular ATP/ADP equilibrium displacement (i.e. $\Delta G_{ATP}$ charge) and thus could potentially result in identical respiratory profiles in intact cells. For example, a small network of mitochondria each respiring near maximal capacity could in theory produce an identical 'basal' oxygen consumption rate to that of a comparatively larger mitochondrial network in which forward respiratory flux was constrained across each mitochondrial unit. Our findings provide definitive support for the latter scenario in AML, as application of our diagnostic biochemical workflow revealed that intrinsic limitations in fractional OXPHOS characterize an expansive mitochondrial network in human leukemia. In fact, a substantial portion of the AML mitochondrial network is incapable of contributing to oxidative ATP production, as leukemic mitochondria primarily consume, rather than produce, ATP across a physiological $\Delta G_{ATP}$ span. Intrinsic OXPHOS limitations in AML appear to derive from a unique biochemical mechanism whereby extra-mitochondrial ATP gains access to the matrix space, where it then directly inhibits electron

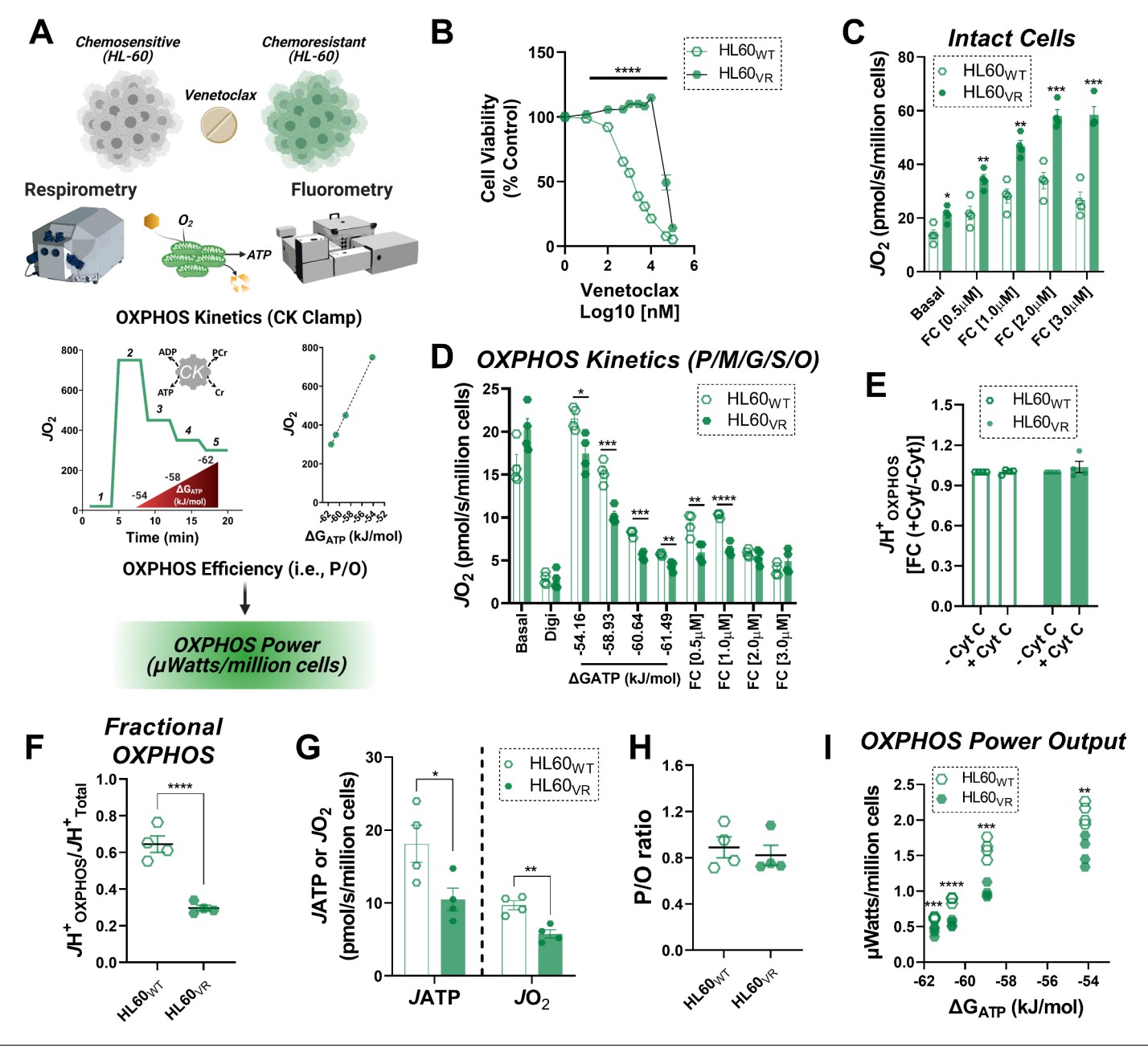

**Figure 8.** OXPHOS power output is reduced in the setting of venetoclax resistance. (A) Study schematic depicting bioenergetic characterization of OXPHOS power output in HL-60 cells either sensitive (HL60WT) or made resistant to venetoclax by continuous exposure to 1 μM venetoclax (HL60VR). (B) Cell viability, expressed as a percentage of control, following 48 hr exposure to increasing concentrations of venetoclax; n = 6 independent experiments. (C) Intact cellular respiration under basal conditions, as well as in response to FCCP titration in bicarbonate free RPMI media. (D) OXPHOS kinetics in permeabilized cells. (E) Effect of cytochrome C addition on JH+OXPHOS. Data expresses as fold change from rates obtained prior to cytochrome C addition. (F) Fractional OXPHOS - note, JH+Total was calculated from the intact cell assay in C using the maximal rate stimulated by FCCP. (G) Rate of ATP synthesis (JATP) and oxygen consumption (JO2) in permeabilized cells energized with P/M/G/S/O. (H) Calculated P/O ratio. (I) Calculated OXPHOS power output. (C–I) n = 4 independent experiments. Data are presented as mean ± SEM analyzed by unpaired t-tests. *p<0.05, **p<0.01, ***p<0.001, ****p<0.0001.

The online version of this article includes the following source data for figure 8:

**Source data 1.** Raw values for '*Figure 8*'.

transport flux in a $\Delta G_{ATP}$-dependent manner. Such inhibition is independent of ATP synthase (i.e. CV) and presumably localized to the respiratory complexes downstream of the ubiquinone pool (i.e. CIII, Cyt C, CIV). Given that evidence for this effect was also observed in bone marrow-derived CD34+ stem cells, allosteric and/or post-translation regulation of ETS flux is likely a primary mode of OXPHOS regulation in hematopoietic progenitors that is maintained during leukemogenesis. Importantly, reversal of this effect was strongly cytotoxic to AML, indicating that direct OXPHOS regulation by $\Delta G_{ATP}$ confers a survival advantage during hematopoietic clonal cell expansion. Although additional work will be required to fully elucidate the mechanism(s) by which ATP uptake directly inhibits OXPHOS flux in AML, our preliminary findings leveraging gamitrinib and curcumin provide proof-of-principle that such regulation can indeed be targeted therapeutically.

While prior work has implicated ANT2 (*Chevrollier et al., 2011*; *Lee et al., 2019*; *Maldonado et al., 2016*) in cancer biology, we present for the first time here a potential mechanism whereby ANT2/3 regulate leukemic cell metabolism via $\Delta G_{ATP}$. Specifically, our findings inform a model of leukemia bioenergetics in which decreased ANT1, and increased ANT2/3 favor the uptake of extra-mitochondrial ATP into the matrix space. The transfer of $\Delta G_{ATP}$ from the cytosol to the matrix in turn constrains OXPHOS via direct inhibition of the ETS. Direct ETS inhibition by $\Delta G_{ATP}$ could derive from multiple sources, such as adenylate-mediated allosteric regulation (*Kadenbach, 2021*), inhibitory phosphorylation of the respiratory complexes (*Helling et al., 2012*), or chaperone mediated protein-protein interactions (*Sciacovelli et al., 2013*). In either case, the exquisite sensitivity of the ETS to $\Delta G_{ATP}$ likely prevents the mitochondrial network from contributing to the cellular $\Delta G_{ATP}$ charge in the leukemic blasts. Consequentially, maintenance of a low cellular $\Delta G_{ATP}$ would be predicted to maximize both glycolytic and mitochondrial metabolism, entirely consistent with the known hyper-metabolic phenotype of human leukemia (*Goto et al., 2014a*; *Goto et al., 2014b*; *Suganuma et al., 2010*). Such conditions are likely advantageous to proliferating leukemic blasts, as insufficient ATPase activity has been demonstrated to be rate-limiting for cellular proliferation (*Luengo et al., 2020*).

Currently, there is great interest in developing novel pharmacotherapies that target mitochondrial OXPHOS in leukemia (*Farge et al., 2017*; *Guièze et al., 2019*; *Kuntz et al., 2017*; *Lee et al., 2015*; *Panina et al., 2020*). Yet, a large caveat of targeting mitochondria is the ubiquitous necessity of OXPHOS for healthy cellular metabolism. If indeed the proliferative potential of leukemia depends upon mitochondrial ATP consumption, rather than production, it is tempting to speculate that a pharmaceutical intervention designed to restore OXPHOS kinetics and/or $\Delta G_{ATP}$ could effectively halt cell proliferation, in turn allowing for proliferating blasts to succumb to apoptosis. Such a targeted approach would be expected to minimize secondary toxicity as increased OXPHOS efficiency is likely advantageous across non-cancerous, highly metabolic tissues (e.g. brain, heart, muscle), as well as in the context of adaptive cellular immunity (*Buck et al., 2016*; *Vardhana et al., 2020*). Based upon our proposed model, targeting the acute OXPHOS regulating capabilities of $\Delta G_{ATP}$ through the VDAC-ANT axis provides an appealing, leukemia-specific target for pharmaceutical intervention, as small-molecule interference of this pathway restored OXPHOS kinetics in leukemic mitochondria and induced cytotoxicity (*Figure 6*). Although prior work has identified TRAP1 (*Bryant et al., 2017*; *Ramkumar et al., 2020*) as a potential anticancer target in leukemia, genetic knockdown in MV-4–11 cells increased, rather than decreased, cellular proliferation in the present study. Of note, the mitochondrial-localized TRAP1 inhibitor gamitrinib is currently being evaluated in a first-in-human phase I clinical trial for treatment of advanced cancers (ClinicalTrials.gov Identifier: NCT04827810). It is currently unclear how much of the cytotoxicity induced by 17-AAG and/or gamitrinib is attributable to on-target TRAP1 inhibition vs off-target interference with ATP exchange via VDAC-ANT. Moreover, the mechanism of cytotoxicity induced by 17-AAG/gamitrinib/curcumin requires additional experimentation. Although speculative, it is possible that the ability of these compounds to selectively block matrix ATP uptake acutely restores full respiratory competence of the expansive AML mitochondrial network, in turn increasing redox pressure across the ETS and boosting ROS production above a cytotoxic threshold.

Taken together, the present findings provide novel insight into the hyper-metabolic phenotype characteristic of human leukemia. With respect to the mitochondria, it bears repeating that heightened basal respiration in AML is almost exclusively assumed to reflect an increased reliance on oxidative ATP synthesis (i.e. OXPHOS). Although this assumption largely underscores the growing interest in targeting mitochondrial oxidative metabolism in cancer, direct interrogation of the

mitochondrial network herein consistently revealed active OXPHOS repression in AML. Representative of this phenotype, TRAP1 knockdown and venetoclax resistance both resulted in an increase in basal respiration that was subsequently linked to decreased, not increased, OXPHOS kinetics. Such findings highlight the diagnostic limitations of metabolic measurements made in intact cells and suggest that too much reliance on binary readouts of respiration versus glycolysis may in fact be masking 'actionable' cancer-specific mitochondrial biology. By leveraging comprehensive mitochondrial diagnostics, our collective findings inform a model whereby intrinsic OXPHOS limitations support aggressive disease dissemination in leukemia and raise the intriguing possibility that pharmaceutical interventions aimed at blocking mitochondrial ATP uptake/consumption warrants further exploration. Given that increased OXPHOS efficiency is advantageous across non-cancerous tissues, a potential benefit to this novel treatment paradigm is the minimization of secondary toxicity (i.e. a wide therapeutic window).

# Materials and methods

**Key resources table**

| Reagent type (species) or resource | Designation | Source or reference | Identifiers | Additional information |
|---|---|---|---|---|
| Biological sample (*Homo sapiens*) | Peripheral Blood Mononuclear Cells; PBMC | Venous puncture | | Freshly isolated from *Homo sapiens*, male and female, 18–75 years |
| Biological sample (*Homo sapiens*) | Bone Marrow Aspirate; Primary Leukemia | Posterior Iliac Crest | | Freshly isolated from *Homo sapiens* |
| Biological sample (*Homo sapiens*) | Human Myoblast; HMB | Muscle biopsies from gastrocnemius muscle | | Freshly isolated from *Homo sapiens* |
| Biological sample (*Homo sapiens*) | CD34+ Cells | HemaCare | CAT#: BM34C | Isolated freshly from bone marrow (Homo sapien, M) |
| Biological sample (*Homo sapiens*) | CD34+ Cells | HemaCare | CAT#: BM34C | Isolated freshly from bone marrow (Homo sapien, M) |
| Biological sample (*Homo sapiens*) | CD34+ Cells | HemaCare | CAT#: BM34C | Isolated freshly from bone marrow (Homo sapien, F) |
| Biological sample (*Homo sapiens*) | Bone Marrow Aspirate; BM Healthy | HemaCare | CAT#: BM008F | Isolated freshly from bone marrow (Homo sapien, F) |
| Biological sample (*Homo sapiens*) | Bone Marrow Aspirate; BM Healthy | HemaCare | CAT#: BM008F | Isolated fresh (Homo sapien, M) |
| Biological sample (*Homo sapiens*) | Bone Marrow Aspirate; BM Healthy | HemaCare | CAT#: BM008F | Isolated fresh (Homo sapien, M) |
| Cell line (*Homo sapien*) | HL-60 | ATCC | CAT#: CCL-240 | |
| Cell line (*Homo sapien*) | MV-4–11 | ATCC | CAT#: CRL-9591 | |
| Cell line (*Homo sapien*) | KG-1 | ATCC | CAT#: CCL-246 | |
| Chemical compound, drug | Oligomycin; Oligo | Tocris | CAT#: 4110 | |
| Chemical compound, drug | FCCP | Millipore Sigma | CAT#: C2920 | |
| Chemical compound, drug | Rotenone; Rot | Millipore Sigma | CAT#: R8875 | |
| Chemical compound, drug | Antimycin; Ant | Millipore Sigma | CAT#: A8674 | |
| Chemical compound, drug | Digitonin; Digi | Millipore Sigma | CAT#: D1410599 | |
| Chemical compound, drug | Potassium Pyruvate; P; Pyr | Combi-Blocks | CAT#: QA-1116 | |

*Continued on next page*

*Continued*

| Reagent type (species) or resource | Designation | Source or reference | Identifiers | Additional information |
|---|---|---|---|---|
| Chemical compound, drug | L-Malic Acid; Malate; M | Alpha Aesar | CAT#: A13702 | |
| Chemical compound, drug | L-Glutamic Acid; Glutamate; G | RPI | CAT#: G25200 | |
| Chemical compound, drug | Succinic Acid; Succinate; S; Succ | Fisher | CAT#: BP336 | |
| Chemical compound, drug | Octanoyl-L-Carnitine; O | Millipore Sigma | CAT#: 50892 | |
| Chemical compound, drug | Cytochrome C | Millipore Sigma | CAT#: C2506 | |
| Chemical compound, drug | ATP | Ark Pharma | CAT#: AK54737 | |
| Chemical compound, drug | Creatine Kinase | Millipore Sigma | CAT#: C2506 | |
| Chemical compound, drug | Phosphocreatine; PCr | Millipore Sigma | CAT#: 237911 | |
| Chemical compound, drug | Carboxy Atractyloside; CAT | Millipore Sigma | CAT#: C4992 | |
| Chemical compound, drug | 17-AAG | Millipore Sigma | CAT#:100068 | |
| Chemical compound, drug | Gamitrinib TPP hexafluorophosphate; Gamitrinib | MedChem Express | CAT#: HY-102007A | |
| Chemical compound, drug | Curcumin | Millipore Sigma | CAT#: 239802 | |
| Chemical compound, drug | Venetoclax | Selleckchem | CAT#: S8048 | |
| Chemical compound, drug | Bongkrekic acid | Cayman Chemical | CAT#: 19079 | |
| Chemical compound, drug | ACK Lysis Buffer | Lonza | CAT#: BP10-548E | |
| Chemical compound, drug | Mitotracker Green-FM Dye; MTG-FM | Thermo Fisher | CAT#: M7514 | |
| Chemical compound, drug | Tetramethyl rhodamine methyl ester (TMRM) | Invitrogen | CAT#: 1924079 | |
| Chemical compound, drug | Pierce Lys-C Protease | Thermo Fisher | CAT#: 90307 | |
| Transfected construct (*Homo sapien*) | shRNA to TRAP1 | Origene | CAT#: TL300868V | shRNA lentiviral particles packaged from pGFP-C-shLenti vector |
| Sequence-based reagent | Primer for TRAP1 | Thermo Fisher | CAT#: 4331182 | |
| Sequence-based reagent | Primer for 18 s rRNA | Thermo Fisher | CAT#: 4319413E | |
| Peptide, recombinant protein | Seq Grade Trypsin | Millipore Sigma | CAT#: V5113 | |
| Peptide, recombinant protein | Human Stem Cell Factor; SCF | Gibco | CAT#: PHC2115 | |
| Peptide, recombinant protein | Human Thrombopoietin | Gibco | CAT#: PHC9514 | |
| Peptide, recombinant protein | FLT-3 Ligand | Sigma Aldrich | CAT#: SRP3044 | |
| Commercial assay or kit | RNeasy Midi kit | Qiagen | CAT#: 74124 | |
| Commercial assay or kit | Superscript IV reverse transcriptase | Invitrogen | CAT#: 18090010 | |

*Continued on next page*

*Continued*

| Reagent type (species) or resource | Designation | Source or reference | Identifiers | Additional information |
|---|---|---|---|---|
| Commercial assay or kit | TMT 10-plex | Thermo Fisher | CAT#: A34808 | |
| Software, algorithm | Prism 8.4 | GraphPad | RRID:SCR_002798 | |
| Software, algorithm | Proteome Discoverer 2.2 | Thermo Fisher | | |
| Software, algorithm | Mito Carta 2.0 | | RRID:SCR_018165 | |

### Ethical approval of human subject research

All procedures involving human subjects were approved by the Institutional Review Board of the Brody School of Medicine at East Carolina University (study ID: UMCIRB 18–001328, UMCIRB 19–002331).

### Blood collection and isolation of PBMCs

For PBMC samples, healthy subjects (ages 18–70 years), without a prior history of hematological malignancy, were recruited from the surrounding area. Following informed consent (study ID: UMCIRB 18–001328), venous blood from the brachial region of the upper arm was collected. Whole blood was collected in sodium-heparinized Cell Preparation Tubes (CPT) (BD Biosciences, Franklin Lakes, NJ) and centrifuged at 1800 x g for 15 min. Prior to experimentation, mononuclear cells were washed in ammonium-chloride-potassium (ACK) lysis buffer to remove red blood cells and either used immediately for experiments or cultured overnight in IMDM (Thermo Fisher Scientific, Waltham, MA) supplemented with glutamine, 10% FBS, and 1% penicillin/streptomycin.

### Mononuclear cell isolation from bone marrow aspirates

For primary leukemia samples, bone marrow aspirates were collected from patients undergoing confirmatory diagnosis for a range of hematological malignancies as a component of an already scheduled procedure. All patients provided informed consent prior to study enrollment (study ID: UMCIRB 19–002331). Type of leukemia ranged from acute myeloid leukemia (AML, N = 14), chronic myeloid leukemia (CML, N = 2), and granular lymphocytic leukemia (N = 1). Patient age ranged from 32 to 78 years (male/female, 8/9). Bone marrow aspirates were collected in sodium-heparinized Cell Preparation Tubes (CPT) (BD Biosciences, Franklin Lakes, NJ) and centrifuged at 1800 x g for 15 min. Mononuclear cells were isolated and then washed in ACK lysis buffer to remove red blood cells and used immediately for experiments. In addition to primary leukemia samples, bone marrow aspirates were collected from healthy donors, ages 26–33, supplied by HemaCare (Northridge, CA).

### CD34+ stem/progenitor cell collection and culture

CD34+ cells were purchased from HemaCare (Northridge, CA). These cells were isolated and purified from bone marrow aspirates of healthy donors, ages 20–39. A portion from each vial was cultured in IMDM supplemented with 15% HyClone FBS, recombinant human SCF (25 ng/mL), TPO (50 ng/mL), and FLT-3 (50 ng/mL).

### AML cell line culture

HL-60, KG-1, and MV-4–11 (ATCC, Manassas, VA) human leukemia cells were cultured in IMDM (Thermo Fisher Scientific, Waltham, MA) supplemented with glutamax, 10% FBS, and 1% penicillin/streptomycin and incubated at 37°C in 5% $CO_2$. All cell lines were obtained from ATCC (Manassas, VA). Cell lines were not tested or authenticated over and above documentation provided by ATCC, which included antigen expression, DNA profile, short tandem repeat profiling, and cytogenetic analysis. For experiments involving venetoclax resistance, HL-60 cells were cultured in RPMI-1640, supplemented with glutamax, 10% FBS, and 1% penicillin/streptomycin. To model venetoclax resistance, HL-60 cells were conditionally adapted over time to 1 µM venetoclax in a manner previously described for other chemotherapy drugs (*Kao et al., 2019*). Upon reaching an average cell density

of $1.5 \times 10^6$ cell/mL the cells were harvested and used for whole cell, permeabilized cell, and isolated mitochondria experiments. Primary human muscle progenitor cells (human myoblasts, 'HMB') were derived from fresh muscle biopsy samples, as described previously (*Ryan et al., 2018*). Cells were cultured on collagen-coated flasks using HMB growth medium (GM: Ham's F10, supplemented with 20% FBS and 1% penicillin/streptomycin, and supplemented immediately prior to use with 5 ng/ml basic FGF).

## Cell viability assays

Cell viability was determined by viable cell count using Trypan Blue (0.4%). Where indicated, viability was determined by fluorescence measurement as previously described (L.-P. *Kao et al., 2019*). Briefly, cells were seeded in black-wall, 96-well plates, in growth medium. After addition of agents (0.1 ml final well volume), cells were incubated at 37° C, 5% $CO_2$, for the times indicated. Viability was determined using propidium iodide (PI) as follows. Positive control cells were permeabilized by addition of 10 µl of 1.0 mg/ml digitonin and incubated at 37° C, 5% $CO_2$, for 20 min. Plates were then centrifuged at 1200 x *g* for 20 min, and after dumping the media, 0.1 ml of a 5.0 µM PI solution in PBS was added. The plate was again incubated for 20 min, and viability was calculated as the mean fluorescence (minus permeabilized vehicle control) at 530 nm excitation and 620 nm emission. For venetoclax-induced cell death assays, cell viability was determined using a standard MTT assay and absorption was read at 570 nm. For all viability assays, each biological replicate was derived from the mean of three technical replicates.

## TRAP1 knockdown in MV-4–11 cells

MV-4–11 cells were cultured in IMDM (Thermo Fisher Scientific, Waltham, MA) supplemented with glutamax, 10% FBS, and 1% penicillin/streptomycin and incubated at 37°C in 5% $CO_2$. Human shRNA lentiviral particles packaged from pGFP-C-shLenti vector (4 unique 29mer TRAP1-specific shRNA [ACAGCCGCAAAGTCCTCATCCAGACCAAG; ATGGTGGCTGACAGAGTGGAGGTCTATTC; GGAGACGGACATAGTCGTGGATCACTACA; TGGCTTTCAGATGGTTCTGGAGTGTTTGA], one scramble control; 0.5 ml each, >107 TU/ml) were purchased from Origene (CAT#: TL300868V). To facilitate infection, MV-4–11 cells and lentiviral particles were co-cultured for 24 hr in individual wells of a 96-well plate in 0.1 mL of IMDM growth media, supplemented with 4 µg/mL polybrene (multiplicity of infection of approximately 20). At the end of the 24 hr, cells were spun down and resuspended in culture media devoid of polybrene. Cultures were then subjected to puromycin selection by continuous exposure to 2 µg/mL puromycin in the culture media. Confirmation of TRAP1 knockdown was performed via real-time PCR. To do this, total RNA was extracted from cell pellets using Qiagen RNeasy Midi kits per manufacturer instructions. RNA was reverse transcribed using Superscript IV reverse transcriptase according to manufacturer instructions (Invitrogen). Real-time PCR on TRAP1 was performed using a Quantstudio 3 Real-Time PCR system (Applied Biosystems). Relative quantification of mRNA levels was determined using the comparative threshold cycle (ΔΔCT) method using FAM-labeled Taqman gene expression assays (Applied Biosystems) specific to TRAP1 run in multiplex with a VIC-labeled 18S control primer.

## Confocal microscopy

Cells were pre-loaded with 200 nM Mitotracker Green-FM dye (MTG-FM; Molecular Probes, Eugene, OR) at 37°C for 1 hr. Cells were then centrifuged at 300 x *g* for 7 min at ~25°C and resuspended in MTG-FM-free IMDM formulation media (Thermo Fisher) containing 50 nM tetramethyl rhodamine methyl ester (TMRM) and 2 µM Hoechst 33342. Cells were plated on glass-bottom dishes (MatTek, Ashland, MA) for imaging. Cells were held in place with a thin 1% agarose pad that was applied immediately prior to imaging in order to minimize rapid motion interference during imaging of live non-adherent cells (*Ivanusic et al., 2017*).

All imaging were performed using an Olympus FV1000 laser scanning confocal microscope (LSCM) with an onstage incubator at 37°C. Acquisition software was Olympus FluoView FSW (V4.2). The objective used was 60X oil immersion (NA = 1.35, Olympus Plan Apochromat UPLSAPO60X(F)). Images were 800 × 800 pixel with 2µs/pixel dwell time, sequential scan mode, resulting in a 4X digital zoom. Hoechst 33342 was excited using the 405 nm line of a multiline argon laser; emission was filtered using a 560 nm dichroic mirror and 420–460 nm barrier filter. MTG-FM was excited using the

488 nm line of a multiline argon laser; emission was filtered using a 560 nm dichroic mirror and 505–540 nm barrier filter. TMRM was excited using a 559 nm laser diode; emission was filtered using a 575–675 nm barrier filter. Zero detector offset was used for all images and gain at the detectors was kept the same for all imaging. The pinhole aperture diameter was set to 105 μm (1 Airy disc).

Images were analyzed using Fiji (*Schindelin et al., 2012*). Spatial resolution was measured using sub-resolution fluorescent beads (Thermo Fisher) and curve fitting was performed using the MetroloJ plugin in Fiji. 16-bit images were made into a composite. Circular ROIs were manually selected using the ROI manager plugin. Images were then decomposed into separate 16-bit image stacks leaving the ROI positions intact. A Huang auto-threshold was used for automated selection of signal for all three channels. Following threshold application, each signal was measured using the multi-measure feature. Only whole cells were analyzed (i.e. cells on edges of the FOV were excluded). Slices containing cells above the lowest monolayer were removed from stacks to avoid oversampling. The following calculations were performed to determine the relevant signal volumes.

$$\text{Signal Volume } (\mu m^3) = [A^*Z]/N$$

where A is the signal-positive area selected using a Huang auto-threshold ($\mu m^2$), Z is the optical section thickness (axial resolution; μm), and N is the number of steps within each optical section (i.e. axial resolution divided by the step size). The latter operation is necessary to correct for oversampling of the signal volumes.

## Respiratory flux in intact and permeabilized cells

Approximately $1–3 \times 10^6$ cells were used for each intact and permeabilized cell experiment. High-resolution respirometry measurements were performed using the Oroboros Oxgraph-2k (O2k; Oroboros Instruments, Innsbruck, Austria) in a 0.5 or 1.0 mL reaction volume at 37°C. For normalization to total protein, at the conclusion of each experiment the cell suspension was collected from each chamber and centrifuged at 2000 x g for 10 min at 4°C. Cells were lysed using low-percentage detergent buffer (CelLytic M) followed by a freeze-thaw cycle, and protein concentration was determined using a BCA protein assay.

Respiratory flux was measured using previously described methods (*Fisher-Wellman et al., 2018*). For intact cell measurements, cells were resuspended in Intact Cell Respiratory Media (17.7 g/L Iscove's Modified Dulbecco's Medium (IMDM), 20 mM HEPES, 1% Penicillin/Streptomycin, 10% FBS, pH 7.4). After basal respiration was established, oligomycin (Oligo; 0.02 μM) was added followed by FCCP titration (FC; 0.5–5 μM), rotenone (Rot; 0.5 μM) and antimycin A (Ant; 0.5 μM). For permeabilized cell measurements, cells were resuspended in Respiratory Buffer supplemented with creatine (105 mM MES potassium salt, 30 mM KCl, 8 mM NaCl, 1 mM EGTA, 10 mM $KH_2PO_4$, 5 mM $MgCl_2$, 0.25% BSA, 5 mM creatine monohydrate, pH 7.2). Cells were permeabilized with digitonin (Digi; 0.02 mg/mL), and respiratory flux was measured using the creatine kinase (CK) clamp and FCCP titration assays. Within the CK clamp assay, the free energy of ATP hydrolysis ($\Delta G_{ATP}$) is calculated using the equilibrium constant for the CK reaction ($K'_{CK}$) and is based upon the addition of known concentrations of creatine (CR), phosphocreatine (PCR), and ATP in the presence of excess amounts of CK (*Fisher-Wellman et al., 2018*). Calculation of $\Delta G_{ATP}$ at defined PCR concentrations was done using the online resource (https://dmpio.github.io/bioenergetic-calculators/ck_clamp/) previously described (*Fisher-Wellman et al., 2018*).

For all assays, various combinations of carbon substrates and inhibitors were employed. Substrates and inhibitors utilized are indicated in the figure legends: CK (20 U/mL), ATP (5 mM), PCR (1 mM, 6 mM, 15 mM, 21 mM), pyruvate (Pyr or P; 5 mM), malate (M; 1 mM), glutamate (G; 5 mM), octanoyl-carnitine (O; 0.2 mM), succinate (Succ or S; 5 mM) cytochrome C (Cyt C, 10 μM), oligomycin (Oligo, 0.02 μM), FCCP (FC, 0.5–2 μM), rotenone (Rot, 0.5 μM), antimycin A (Ant A, 0.5 μM), carboxyatractyloside (CAT, 1 μM), bongkrekic acid (20 μM), 17-AAG (sigma, #100068, 1 μM), Gamitrinib TPP hexafluorophosphate (MedChemExpress, #HY-102007A, 1 μM).

## Extracellular flux analysis

Extracellular acidification rate (ECAR) and oxygen consumption rate (OCR) were measured using a Seahorse XF24e flux analyzer (Agilent Technologies, Santa Clara, CA). Prior to analysis, wells were coated with Cell-Tak (Corning, Cat# 354240). Cells were then seeded at $3 \times 10^5$ cells/well. The assay

was performed in bicarbonate free IMDM, supplemented with 2 mM HEPES, in the absence of FBS, pH 7.4. The flux analysis protocol was as follows: Basal OCR and ECAR were measured followed by the addition of 5 µM Oligomycin, and 50 mM 2-deoxyglucose (2-DG). All data were normalized to viable cell count.

## Isolation of mitochondria from PBMCs and leukemia cells

In order to pellet cells, PBMC fractions were washed with PBS and centrifuged at 300 x g for 10 min at 4°C and leukemia cells were centrifuged at 300 x g for 10 min followed by a PBS wash. Cell pellets were resuspended in Mitochondrial Isolation Buffer with BSA (100 mM KCl, 50 mM MOPS, 1 mM EGTA, 5 mM MgSO$_4$, 0.2% BSA, pH 7.1) and homogenized using a borosilicate glass mortar and Teflon pestle. Homogenates were centrifuged at 800 x g for 10 min at 4°C. The supernatant was collected, and the remaining pellet was resuspended in Mitochondrial Isolation Buffer with BSA, then homogenized and centrifuged again. This process was repeated a total of three times. The collected supernatant was centrifuged at 10,000 x g for 10 min at 4°C to pellet the mitochondrial fraction. The fraction was resuspended in Mitochondrial Isolation Buffer without BSA, transferred to a microcentrifuge tube and subjected to another spin at 10,000 x g. The mitochondrial pellet was resuspended in ~100 µL of Mitochondria Isolation Buffer and protein concentration was calculated using the Pierce BCA assay. Respiration assays using isolated mitochondria were similar to that described for permeabilized cells.

## Mitochondrial NADH/NAD$^+$ redox in isolated mitochondria

Fluorescent determination of NADH/NAD$^+$ was performed using a QuantaMaster Spectrofluorometer (QM-400, Horiba Scientific, Kyoto, Japan). The NADH/NAD + was detected at Ex/Em: 350/450. NADH/NAD + was measured in mitochondria isolated from PBMC and leukemia cell lines using the CK clamp assay. Experiments were performed at 37°C in a 200 µL reaction volume. To start, Respiratory Buffer supplemented with creatine (200 µL), Cyt C (10 µM), mitochondrial lysate (100 µg) were added into a glass cuvette. Mitochondria were incubated at 37°C for ~5 min in the absence of substrate to induce 0% reduction of the NADH pool. Saturating carbon substrates were added (P/M/G/S/O), and respiration was stimulated with the CK clamp. Titration of $\Delta G_{ATP}$ was performed via PCR titration (6, 15, 21 mM). Oligomycin (0.02 µM) was added to inhibit ATP synthesis and cyanide (CN, 10 mM) was added to induce 100% reduction of the matrix NADH pool. The NADH/NAD + was expressed as a percentage reduction of the CN value (i.e. 100% reduction) based upon the formula % Reduction = $(F-F_{0\%})/(F_{100\%}-F_{0\%})*100$.

## Mitochondrial membrane potential ($\Delta\Psi$) in isolated mitochondria

Fluorescent determination of $\Delta\Psi$ was carried out via a QuantaMaster Spectrofluorometer (QM-400; Horiba Scientific). Determination of $\Delta\Psi$ via TMRM was done as described previously (*Fisher-Wellman et al., 2018*), taking the fluorescence ratio of the following excitation/emission parameters [Ex/Em, 576/590 and 551/590]. The 576/551 ratio was then converted to millivolts via a KCl standard curve performed in the presence of valinomycin as described (*Fisher-Wellman et al., 2018*).

## Mitochondrial JATP synthesis, P/O ratio and OXPHOS power output in permeabilized cells

Parallel respiration and fluorometric experiments were carried out in order to generate an ATP/O (P/O) ratio, as previously described (*Lark et al., 2016*). Experiments were performed at 37°C in a 2.5 mL reaction volume in Respiratory Buffer supplemented with AP5A (0.2 mM), and digitonin (0.02 mg/mL). ATP synthesis and oxygen consumption were determined in permeabilized cells energized with P/M/G/S/O in the presence of 0.1 mM ADP. The P/O ratio was then calculated using the ratio of the JATP and the steady state $JO_2$, divided by 2. Empirically derived P/O ratios were then used to convert oxygen consumption rates recorded during the CK clamp assay (using identical substrates) to ATP production rate [(oxygen consumption rate) x (P/O x 2)]. A fixed extra-mitochondrial force was assumed to be applied via the CK clamp at each PCR titration. These forces corresponded to −54.16, −58.93, −60.64, and −61.49 kJ/mol. These forces were used along with ATP production rate to quantitate OXPHOS power output in Watts (J/s) according to the following formula [(pmol ATP/s/million cells) x (5.416 or 5.893 or 6.064 or 6.149 * 10$^{-8}$ J/pmol ATP) = µWatts/million cells].

## Mitochondrial lysis, protein digestion, and peptide labeling for TMT quantitative proteomics

Mitochondrial pellets from leukemia cells and PBMC (approximately 250 µg of protein) were lysed in ice-cold 8 M Urea Lysis Buffer (8 M urea in 50 mM Tris, pH 8.0, 40 mM NaCl, 2 mM CaCl$_2$, 1x cOmplete ULTRA mini EDTA-free protease inhibitor tablet), as described previously (*McLaughlin et al., 2020*). The samples were frozen on dry ice and thawed for three freeze-thaw cycles and further disrupted by sonication with a probe sonicator in three 5 s bursts set at an amplitude of 30 (Q Sonica, Newtown, CT). Samples were centrifuged at 10,000 × g for 10 min at 4°C to pellet insoluble material. Protein concentration was determined by BCA, and equal amounts of protein (200 µg, adjusted to 2.5 mg/mL with Urea Lysis Buffer) were reduced with 5 mM DTT at 32°C for 30 min, cooled to room temperature, and then alkylated with 15 mM iodoacetamide for 30 min in the dark. Unreacted iodoacetamide was quenched by the addition of DTT up to 15 mM. Initial digestion was performed with Lys C (Thermo Fisher) 1:100 w-w; 2 µg enzyme per 200 µg protein for 4 hr at 32°C. Following dilution to 1.5 M urea with 50 mM Tris (pH 8.0), 30 mM NaCl, 5 mM CaCl$_2$, the samples were digested overnight with trypsin (Promega, Madison, WI) 50:1 w/w, protein:enzyme at 32°C. Samples were acidified to 0.5% TFA and centrifuged at 10,000 × g for 10 min at 4°C to pellet insoluble material. Supernatant containing soluble peptides was desalted on a 50 mg tC18 SEP-PAK solid phase extraction column (Waters, Milford, MA) and eluted (500 µL 25% acetonitrile/0.1% TFA and 2 × 500 µL 50% acetonitrile/0.1% TFA). The 1.5 mL eluate was frozen and lyophilized.

## TMT labeling

TMT labeling was performed as previously described (*McLaughlin et al., 2020*). The samples from isolated mitochondria were re-suspended in 100 µL of 200 mM triethylammonium bicarbonate (TEAB), mixed with a unique 10-plex Tandem Mass Tag (TMT) reagent (0.8 mg re-suspended in 50 µL100% acetonitrile), and shaken for 4 hr at room temperature (Thermo Fisher). A total of 2 × 10 plex kits were used and one sample was TMT-labeled in both kits to control for quantification differences across multiplex preparations. Following quenching with 0.8 µL 50% hydroxylamine samples were frozen, and lyophilized. Samples were re-suspended in ~1 mL of 0.5% TFA and again subjected to solid phase extraction, but with a 100 mg tC18 SEP-PAK SPE column (Waters). The multiplexed peptide sample was subjected to high pH reversed phase fractionation according to the manufacturer's instructions (Thermo Fisher). In this protocol, peptides (100 µg) are loaded onto a pH-resistant resin and then desalted with water washing combined with low-speed centrifugation. A step-gradient of increasing acetonitrile concentration in a high-pH elution solution is then applied to columns to elute bound peptides into eight fractions. Following elution, fractions were frozen and lyophilized.

nLC-MS/MS for TMT proteomics nLC-MS/MS was performed as described previously (*McLaughlin et al., 2020*). Peptide fractions were suspended in 0.1% formic acid at a concentration of 0.25 µg/µL, following peptide quantification (ThermoFisher). All samples were subjected to nanoLC-MS/MS analysis using an UltiMate 3000 RSLCnano system (Thermo Fisher) coupled to a Q Exactive PlusHybrid Quadrupole-Orbitrap mass spectrometer (Thermo Fisher) via nanoelectrospray ionization source. For each injection of 4 µL (1 µg), the sample was first trapped on an Acclaim PepMap 100 20 mm ×0.075 mm trapping column (Thermo Fisher) 5 µl/min at 98/2 v/v water/acetonitrile with 0.1% formic acid, after which the analytical separation was performed over a 90-min gradient (flow rate of 300 nanoliters/min) of 3–30% acetonitrile using a 2 µm EASY-Spray PepMap RSLC C18 75 µm × 250 mm column (Thermo Fisher) with a column temperature of 55°C. MS1 was performed at 70,000 resolution, with an AGC target of 1 × 10$^6$ ions and a maximum IT of 60 ms. MS2 spectra were collected by data-dependent acquisition (DDA) of the top 20 most abundant precursor ions with a charge greater than one per MS1 scan, with dynamic exclusion enabled for 30 s. Precursor ions were filtered with a 1.0 m/z isolation window and fragmented with a normalized collision energy of 30. MS2 scans were performed at 17,500 resolution, AGC target of 1 × 10$^5$ ions, and a maximum IT of 60 ms.

## Data analysis for TMT proteomics

Proteome Discoverer 2.2 (PDv2.2) was used for raw data analysis, with default search parameters including oxidation (15.995 Da on M) as a variable modification and carbamidomethyl (57.021 Da on

C) and TMT6plex (229.163 Da on peptide N-term and K) as fixed modifications, and two missed cleavages (full trypsin specificity). Data were searched against human Mito Carta 2.0 database (*Calvo et al., 2016*). PSMs were filtered to a 1% FDR. PSMs were grouped to unique peptides while maintaining a 1% FDR at the peptide level. Peptides were grouped to proteins using the rules of strict parsimony and proteins were filtered to 1% FDR using the Protein FDR Validator node of PD2.2. MS2 reporter ion intensities for all PSMs having co-isolation interference below 0.5 (50% of the ion current in the isolation window) and an average S/N > 10 for reporter ions were summed together at the peptide and protein level. Imputation was performed via low abundance resampling.

### Statistical analysis for TMT proteomic

The protein group tab in the PDv2.2 results was exported as tab delimited.txt. files, and analyzed based on a previously described workflow (*McLaughlin et al., 2020*). First, M2 reporter (TMT) intensities were summed together for each TMT channel, each channel's sum was divided by the average of all channels' sums, resulting in channel-specific loading control normalization factors to correct for any deviation from equal protein input in the 10-plex experiments. Reporter intensities for proteins were divided by the loading control normalization factors for each respective TMT channel. All loading control-normalized reporter intensities were converted to $\log_2$ space and the average value from the 10 samples per kit was subtracted from each sample-specific measurement to normalize the relative measurements to the mean of each kit. Data from each kit were then combined for statistical comparisons. For comparison of PBMC to leukemia cell lines, condition average, standard deviation, p-value (p, two-tailed student's t-test, assuming equal variance), and adjusted p-value ($P_{adjusted}$, Benjamini Hochberg FDR correction) were calculated (*Benjamini and Hochberg, 1995*; *Lesack and Naugler, 2011*). For protein-level quantification, only Master Proteins—or the most statistically significant protein representing a group of parsimonious proteins containing common peptides identified at 1% FDR—were used for quantitative comparison.

### Data availability

All data from the manuscript are available upon request. In addition, all data are available in the source data files provided with this paper. All raw data for proteomics experiments is available online using accession number 'PXD020715' for Proteome Xchange (*Deutsch et al., 2017*) and accession number 'JPST000934' for jPOST Repository (*Okuda et al., 2017*).

### Statistical analysis and software

Statistical analysis was performed using GraphPad Prism 8.4. All data are represented as mean ± SEM and analysis were conducted with a significance level set at p<0.05. Details of statistical analysis are included within figure legends. Figures were generated using Biorender and GraphPad Prism 8.4.

## Acknowledgements

The work was supported by DOD-W81XWH-19-1-0213 (K.H.F.-W.) and NIH P01 CA171983 (M.C.C). This project used the North Carolina Tissue Consortium (NCTC) shared resource which is supported in part by the University Cancer Research Fund (UCRF).

## Additional information

### Funding

| Funder | Grant reference number | Author |
|---|---|---|
| U.S. Army Medical Research and Materiel Command | W81XWH-19-1-0213 | Kelsey H Fisher-Wellman |
| National Cancer Institute | P01 CA171983 | Myles C Cabot |

The funders had no role in study design, data collection and interpretation, or the decision to submit the work for publication.

## Author contributions
Margaret AM Nelson, Kelsey L McLaughlin, Conceptualization, Data curation, Formal analysis, Investigation, Methodology, Writing - original draft, Writing - review and editing; James T Hagen, Conceptualization, Data curation, Investigation, Methodology, Writing - review and editing; Hannah S Coalson, Data curation, Investigation, Writing - review and editing; Cameron Schmidt, Data curation, Formal analysis, Investigation, Methodology, Writing - original draft, Writing - review and editing; Miki Kassai, Investigation; Kimberly A Kew, Data curation, Methodology, Writing - review and editing; Joseph M McClung, Conceptualization, Resources, Methodology, Writing - review and editing; P Darrell Neufer, Conceptualization, Supervision, Writing - review and editing; Patricia Brophy, Resources, Project administration, Writing - review and editing; Nasreen A Vohra, Darla Liles, Conceptualization, Resources, Project administration, Writing - review and editing; Myles C Cabot, Conceptualization, Resources, Funding acquisition, Project administration, Writing - review and editing; Kelsey H Fisher-Wellman, Conceptualization, Resources, Data curation, Software, Formal analysis, Supervision, Funding acquisition, Validation, Investigation, Visualization, Methodology, Writing - original draft, Project administration, Writing - review and editing

## Author ORCIDs
Kelsey H Fisher-Wellman  https://orcid.org/0000-0002-0300-829X

## Ethics
Human subjects: All procedures involving human subjects were approved by the Institutional Review Board of the Brody School of Medicine at East Carolina University (study ID: UMCIRB 18-001328, UMCIRB 19-002331). For PBMC samples, healthy subjects (ages 18-70 years), without a prior history of hematological malignancy, were recruited from the surrounding area. Following informed consent (study ID: UMCIRB 18-001328), venous blood from the brachial region of the upper arm was collected. For primary leukemia samples, bone marrow aspirates were collected from patients undergoing confirmatory diagnosis for a range of hematological malignancies as a component of an already scheduled procedure. All patients provided informed consent prior to study enrollment (study ID: UMCIRB 19-002331).

## Decision letter and Author response
Decision letter https://doi.org/10.7554/eLife.63104.sa1
Author response https://doi.org/10.7554/eLife.63104.sa2

# Additional files
## Supplementary files
• Supplementary file 1. Mitochondrial proteome of AML cell lines, relative to PBMC. (**A**) Exported results from PDv2.2. (**B**) Analyzed master protein expression by group.

• Transparent reporting form

## Data availability
All data from the manuscript are available upon request. In addition, all data are available in the source data files provided with this paper. Raw data for proteomics experiments are available online using accession number "PXD020715" for Proteome Xchange and accession number "JPST000934" for jPOST Repository http://proteomecentral.proteomexchange.org/cgi/GetDataset?ID=PXD020715.

The following datasets were generated:

| Author(s) | Year | Dataset title | Dataset URL | Database and Identifier |
|---|---|---|---|---|
| Fisher-Wellman KH | 2020 | Mitochondrial proteome of human leukemia. | http://proteomecentral.proteomexchange.org/ | Proteome Xchange , PXD020715 |

| | | | cgi/GetDataset?ID=PXD020715 | |
| Fisher-Wellman KH | 2020 | Mitochondrial proteome of human leuekmia | https://repository.jpostdb.org/entry/JPST000934 | jPOST Repository , JPST000934 |

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
