## [Decision Letter]

**Acceptance summary:**

In this article, Nelson et al. carried out systematic biochemical characterization of mitochondrial functions between leukemic cells and normal leukocytes. The authors show that intrinsic limitations in oxidative phosphorylation in leukemic cells distinguish them from normal leukocytes. Evidence is provided that this specific mitochondrial phenotype of leukemic cells is at least in part mediated via the adenine nucleotide translocase (ANT)/ the voltage-dependent anion channel (VDAC) axis and aberrant adenylate transport across the outer mitochondrial membrane. Collectively, it was thought that this study is of broad interest inasmuch as it provides previously unappreciated insights into the molecular underpinnings of the bioenergetic perturbations in leukemia that in long-term may lead to potential therapeutic considerations.

**Decision letter after peer review:**

Thank you for submitting your article "Intrinsic oxidative phosphorylation limitations underlie cellular bioenergetics in leukemia" for consideration by *eLife*. Your article has been reviewed by 3 peer reviewers, including Ivan Topisirovic as the Reviewing Editor and Reviewer #1, and the evaluation has been overseen by Kathryn Cheah as the Senior Editor. The following individual involved in review of your submission has agreed to reveal their identity: Paolo Gallipoli (Reviewer #2).

The reviewers have discussed the reviews with one another and the Reviewing Editor has drafted this decision to help you prepare a revised submission.

Summary:

In this article, Nelson et al. provide evidence suggesting potential differences in functionality of mitochondria between leukemic cells and normal leukocytes. Specifically, the authors performed comparative biochemical characterization of mitochondrial function between normal and leukemic cells which demonstrated that mitochondria of leukemic cells exhibit intrinsic limitations in oxidative phosphorylation (OXPHOS) relative to peripheral blood mononuclear cells (PBMC). The authors go on to provide insights into underpinning mechanisms. They identified ANT and TRAP1 proteins as differentially expressed between PBMCs and leukemic cells, whereby pharmacological inhibition of these factors resulted in partial reversion of mitochondrial phenotypes. Overall, it was found that this study was of broad interest inasmuch as it provides previously unappreciated insights into the bioenergetics of leukemia with potential therapeutic considerations. Nevertheless, several weaknesses were observed related to non-optimal controls and lack of orthogonal genetic validation of the results obtained using pharmacological approaches.

Essential revisions:

1. PBMCs were deemed to be insufficient control since the observed differences between PBMCs and leukemic cells may result from distinct differentiation stages rather than reflecting normal vs. malignant state. Although it was appreciated that human myoblasts were used as an alternative control, it was found that they do not resolve the latter issue as herein the differences could also stem from tissue specificity. It was thus suggested that normal CD34+ hematopoietic cells should be employed as a control.

2. Results using ANT and TRAP1 inhibitors should be validated using appropriate genetic approaches. Manipulation of ANT2 and/or TRAP1 levels and/or activity should also be accompanied by functional assays (e.g. proliferation, survival etc).

3. The authors should consider altering expression and/or activity of ANTs and TRAP1 in normal CD34+ hematopoietic cells and study impact of such interventions on their fate.

4. Testing interactions between TRAP1 inhibitors and standard AML therapies (e.g. cytarabine) would further strengthen this study.

5. Considering the central role of glycolysis in bioenergetics, it was thought that parallel characterization of mitochondrial functions and glycolysis between normal and leukemic cells may be warranted.

---

## [Author Response]

Essential revisions:1. PBMCs were deemed to be insufficient control since the observed differences between PBMCs and leukemic cells may result from distinct differentiation stages rather than reflecting normal vs. malignant state. Although it was appreciated that human myoblasts were used as an alternative control, it was found that they do not resolve the latter issue as herein the differences could also stem from tissue specificity. It was thus suggested that normal CD34+ hematopoietic cells should be employed as a control.

The decision to use PBMC as a control was based on the assumption that comprehensive bioenergetic phenotyping of non-proliferative PBMC compared to various AML cell lines would provide sufficient experimental design contrast to reveal fundamental mitochondrial bioenergetic phenotypes potentially required for clonal cell expansion. Practically speaking, PBMC were also used because they were amendable to discovery science in which multiple assays could be performed for each biological prep. For example, most respirometry experiments required a minimum of 3x10^6^ cells for permeabilized assays and ~150x10^6^ for isolated mitochondria. The use of PBMC allowed us to perform multiple experiments using permeabilized cells, as well as perform assays on isolated mitochondria for each human subject.

Yet, the reviewers’ points regarding alternative controls were well taken especially since the AML cell lines were from different points within the progenitor cell lineage (HL60: promyeloblast, KG1: myeloblast, MV-4-11: lymphoblast). To address this concern, we repeated a subset of key experiments using 3 new control groups. First, as a progenitor/stem cell control we purchased commercially available, purified CD34+ cells. We also used purified CD34+ cells exposed to proliferative stimuli/growth factors (TPO, SLC, FLT-3) as a comparator of clonal expansion. In this respect, unstimulated CD34+ (i.e., CD34+ cells not exposed to proliferative stimuli) control for cell lineage and stimulated CD34+ cells control for AML-phenotypes linked to proliferation that are independent of malignancy. Lastly, mononuclear cells were isolate from bone marrow aspirates obtained from healthy volunteers. One of the major findings with the addition of the new control groups was that relative to naïve CD34+ or PBMC, fractional OXPHOS (proportion of proton current that is used for ATP synthesis) was significantly lower in clonally expanding CD34+ cells (i.e., CD34+_GFs_), as well as primary AML, highlighting that OXPHOS limitations are inherent to, and potentially beneficial for, hematopoietic clonal cell proliferation. Based upon these results, we observed similar bioenergetic phenotypes between PBMC and CD34+, prompting us to feel confident in our use of PBMCs as a control. Results from the additional groups are now displayed in Figure 2 and new Figure 7.

In addition, we also now include data detailing how mitochondrial OXPHOS kinetics are altered in response to acquired chemotherapy resistance (Figure 8). Although upregulations in mitochondrial content and apparent increases in oxidative metabolism are documented features of drug resistance in AML, very little information exists that specifically relates to OXPHOS. We thus investigated fractional OXPHOS and OXPHOS power output in chemosensitive (i.e., ‘wt’) HL-60 cells, as well as HL-60 cells made to be refractory to venetoclax. The use of ‘wt’ counterparts eliminated the need for a ‘non-cancer’ control and thus allowed us to compare OXPHOS kinetics in relation to AML biology in a well-controlled system. As seen when comparing AML to naïve CD34+, basal respiration and maximal respiratory capacity were further elevated in venetoclax resistant HL-60. However, despite having nearly 2-fold more mitochondria per cell than wt, absolute OXPHOS capacity, fractional OXPHOS and OXPHOs power output were not increased, but decreased in venetoclax resistant cells. Together these data support the overall model that indicates that low fractional OXPHOS supports aggressive disease dissemination in AML and thus we feel strongly that these data belong in the current report.

2. Results using ANT and TRAP1 inhibitors should be validated using appropriate genetic approaches. Manipulation of ANT2 and/or TRAP1 levels and/or activity should also be accompanied by functional assays (e.g. proliferation, survival etc).

Based on the reviewers’ suggestions, we prioritized genetic manipulation of TRAP1 as we hypothesized that ΔG_ATP_-mediated ETS inhibition may be driven by acute activation of TRAP1. MV-4-11 cells were infected with lentivirus containing pooled short hairpin RNA (shRNA) against TRAP1 which led to a > 90% reduction in TRAP1 mRNA. Control cells were infected with lentivirus containing scrambled shRNA. All constructs encoded GFP, as well as a puromycin selection gene to allow for stable selection.

Consistent with previous reports, we noticed an increase in basal respiration in TRAP1 knockdown cells. But surprisingly, and opposite to that observed with acute use of TRAP1 inhibitors (17-AAG/gamitrinib), TRAP1 knockdown exacerbated the ability of ATP free energy to blunt ETS flux. These findings clearly indicate that the ability of 17-AAG/gamitrinib to prevent ΔG_ATP_-mediated ETS inhibition is independent of TRAP1 and thus attributable to an ‘off-target’ mechanism. Given that prior research had demonstrated an ability of 17-AAG to bind VDAC, coupled with the fact that a different molecule, curcumin, essentially phenocopied the effect of 17-AAG, the TRAP1 KD experiments informed a new model to explain low fractional OXPHOS in AML. Prior work has demonstrated that curcumin-VDAC interaction locks VDAC in the ‘closed’ confirmation, in turn making it partially selective for cation (i.e., ADP^3-^), rather than anion (i.e., ATP^4-^), transport. Based on this, our biochemical data inform a model in which 17-AAG, gamitrinib and curcumin are capable of selectively blocking matrix ATP uptake, while remaining permissible to ADP to support oxidative ATP synthesis (i.e., OXPHOS). Results from these experiments are now displayed in new Figure 6. We provide evidence of 3 small molecules that appear capable of ‘reversing’ this effect in vitro. Importantly, all 3 compounds induce cytotoxicity in AML, suggesting that restoring OXPHOS may impart therapeutic efficacy in AML. In either case, the transfer of ΔG_ATP_ from the cytosol to the matrix represents an appealing, leukemia specific actionable mitochondrial phenotype that deserves additional consideration. With respect to the mechanism of cytotoxicity, although speculative, it is possible that selectively targeting matrix ATP uptake in leukemic blasts restores full respiratory competence of the already expansive mitochondrial network. This may in turn increase redox pressure across the ETS and boost ROS production above a cytotoxic threshold.

3. The authors should consider altering expression and/or activity of ANTs and TRAP1 in normal CD34+ hematopoietic cells and study impact of such interventions on their fate.

Although we were able to explore the impact of TRAP1 knockdown in MV411 cells, infection efficiency was quite low in these cells and thus required repeated rounds of puromycin selection to generate polyclonal populations of cells in which TRAP1 expression was >90% depleted. Based on this, we did not feel it was feasible to explore genetic manipulations in CD34+ cells as suggested. However, we did investigate the impact of CAT, 17-AAG, and gamitrinib on OXPHOS kinetics in CD34+ (naïve and stimulated to proliferate), bone marrow mononuclear cells from healthy subjects and PBMC. In addition, we assessed cell viability in response to 24hr exposure to 17-AAG or gamitrinib in PBMC. With respect to OXPHOS kinetics, neither CAT, 17-AAG, or gamitrinib impacted OXPHOS kinetics in any of the non-cancer control groups. The notable exception was the CD34+ stimulated to undergo clonal cell expansion. Based on all the data, we interpret these findings to indicate that mitochondrial ATP uptake/consumption is a unique bioenergetic feature that tracks with hematopoietic clonal cell expansion. Interestingly, similar OXPHOS phenotypes are apparent in AML, as well as clonally expanding CD34+, indicating that mitochondrial ATP consumption is not exclusive to malignant growth.

4. Testing interactions between TRAP1 inhibitors and standard AML therapies (e.g. cytarabine) would further strengthen this study.

This is a valid suggestion and previous studies have shown synergistic effects on cell toxicity with sequential treatment of nucleoside analogs and 17-AAG (PMID: 15784732, 14570880). To address this, TRAP1 knockdown and control (scrambled shRNA) cells were exposed for 24 hours to gamitrinib (mitochondrial targeted TRAP1 inhibitor) or curcumin (TRAP1 and VDAC inhibitor), either alone or in combination with cytarabine (Ara-C). Compared to marginal cytotoxicity in response to Ara-C dose escalation, 24hr exposure to either gamitrinib or curcumin decreased cell viability by as much as 60%. Additionally, PBMC were exposed for 24hr to 17-AAG or gamitrinib in the presence of Ara-C, and neither combination impacted cell viability. These findings suggests that gamitrinib and curcumin extend greater cytotoxicity compared to conventional AML therapy and provide proof-of-principle that restoring, rather than disrupting, OXPHOS may have therapeutic potential for treating blood cancer. These findings are presented in the new Figure 6 and Figure 7—figure supplement 1.

5. Considering the central role of glycolysis in bioenergetics, it was thought that parallel characterization of mitochondrial functions and glycolysis between normal and leukemic cells may be warranted.

To address the reviewers’ suggestion, measurements of extracellular acidification (ECAR), an indirect readout of glycolytic flux, were measured. Results revealed higher ECAR and a rightward shift in ECAR relative to oxygen consumption rate (OCR) in AML cell lines (Supp. Figure 1E). These findings were consistent with prior reports detailing a hyper-metabolic phenotype in AML.